# HYPERPARAMETER TRANSFER ACROSS DEVELOPER ADJUSTMENTS

## ABSTRACT

After developer adjustments to a machine learning (ML) algorithm, how can the results of an old hyperparameter optimization (HPO) automatically be used to speedup a new HPO? This question poses a challenging problem, as developer adjustments can change which hyperparameter settings perform well, or even the hyperparameter search space itself. While many approaches exist that leverage knowledge obtained on previous *tasks*, so far, knowledge from previous *development steps* remains entirely untapped. In this work, we remedy this situation and propose a new research framework: hyperparameter transfer across adjustments (HT-AA). To lay a solid foundation for this research framework, we provide four simple HT-AA baseline algorithms and eight benchmarks changing various aspects of ML algorithms, their hyperparameter search spaces, and the neural architectures used. The best baseline, on average and depending on the budgets for the old and new HPO, reaches a given performance 1.2–3.6x faster than a prominent HPO algorithm without transfer. As HPO is a crucial step in ML development but requires extensive computational resources, this speedup would lead to faster development cycles, lower costs, and reduced environmental impacts. To make these benefits available to ML developers off-the-shelf and to facilitate future research on HT-AA, we provide python packages for our baselines and benchmarks.

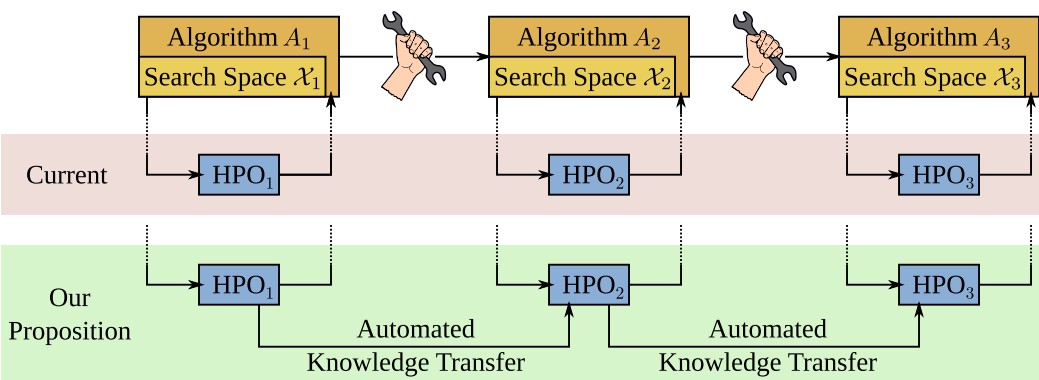

Graphical Abstract: Hyperparameter optimization (HPO) across adjustments to the algorithm or hyperparameter search space. A common practice is to perform HPO from scratch after each adjustment or to somehow manually transfer knowledge. In contrast, we propose a new research framework about automatic knowledge transfers across adjustments for HPO.

## 1 INTRODUCTION: A NEW HYPERPARAMETER TRANSFER FRAMEWORK

The machine learning (ML) community arrived at the current generation of ML algorithms by performing many iterative adjustments. Likely, the way to artificial general intelligence requires many more adjustments. Each algorithm adjustment could change which settings of the algorithm's hyperparameters perform well, or even the hyperparameter search space itself (Chen et al., 2018; Li et al., 2020). For example, when deep learning developers change the optimizer, the learning rate's optimal value

likely changes, and the new optimizer may also introduce new hyperparameters. Since ML algorithms are known to be very sensitive to their hyperparameters (Chen et al., 2018; Feurer & Hutter, 2019), developers are faced with the question of how to adjust their hyperparameters after changing their code. Assuming that the developers have results of one or several hyperparameter optimizations (HPOs) that were performed before the adjustments, they have two options:

1. Somehow manually transfer knowledge from old HPOs.

This is the option chosen by many researchers and developers, explicitly disclosed, e.g., in the seminal work on AlphaGo (Chen et al., 2018). However, this is not a satisfying option since manual decision making is time-consuming, often individually designed, and has already lead to reproducibility problems (Musgrave et al., 2020).

2. Start the new HPO from scratch.

Leaving previous knowledge unutilized can lead to higher computational demands and worse performance (demonstrated empirically in Section 5). This is especially bad as the energy consumption of ML algorithms is already recognized as an environmental problem. For example, deep learning pipelines can have $CO_2$ emissions on the order of magnitude of the emissions of multiple cars for a lifetime (Strubell et al., 2019), and their energy demands are growing furiously: Schwartz et al. (2019) cite a "300,000x increase from 2012 to 2018". Therefore, reducing the number of evaluated hyperparameter settings should be a general goal of the community.

The **main contribution** of this work is the introduction of a new research framework: *Hyperparameter transfer across adjustments (HT-AA)*, which empowers developers with a third option:

3. Automatically transfer knowledge from previous HPOs.

This option leads to advantages in two aspects: The automation of decision making and the utilization of previous knowledge. On the one hand, the automation allows to benchmark strategies, replaces expensive manual decision making, and enables reproducible and comparable experiments; on the other hand, utilizing previous knowledge leads to faster development cycles, lower costs, and reduced environmental impacts.

To lay a solid foundation for the new HT-AA framework, our **individual contributions** are as follows:

- We formally introduce a basic version of the HT-AA problem (Section 2).
- We provide four simple baseline algorithms for our basic HT-AA problem (Section 3).
- We provide a comprehensive set of eight novel benchmarks for our basic HT-AA problem (Section 4).
- We show the advantage of transferring across developer adjustments: some of our simple baseline algorithms outperform HPO from scratch up to 1.2–3.6x on average depending on the budgets (Section 5).
- We empirically demonstrate the need for well-vetted algorithms for HT-AA: two baselines modelled after actually-practiced manual strategies perform horribly on our benchmarks (Section 5).
- We relate the HT-AA framework to existing research efforts and discuss the research opportunities it opens up (Section 6).
- To facilitate future research on HT-AA, we provide open-source code for our experiments and benchmarks and provide a python package with an out-of-the-box usable implementation of our HT-AA algorithms.

## 2  HYPERPARAMETER TRANSFER ACROSS ADJUSTMENTS

After presenting a broad introduction to the topic, we now provide a detailed description of hyperparameter transfer across developer adjustments (HT-AA). We first introduce hyperparameter optimization, then discuss the types of developer adjustments, and finally describe the transfer across these adjustments.

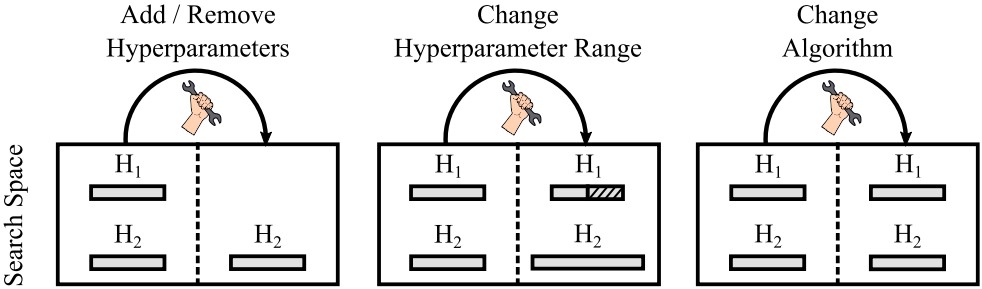

Figure 1: Developer adjustments from the perspective of hyperparameter optimization.

**Hyperparameter optimization (HPO)**   The HPO formulation we utilize in this work is as follows:

$$\underset{\boldsymbol{x}\in\mathcal{X}}{\text{minimize}}\, f_{\mathcal{A}}(\boldsymbol{x}) \quad \text{with } b \text{ evaluations} \qquad , \tag{1}$$

where $f_{\mathcal{A}}(\boldsymbol{x})$ is the objective function for ML algorithm $\mathcal{A}$ with hyperparameter setting $\boldsymbol{x}$, $b$ is the number of available evaluations, and $\mathcal{X}$ is the search space. We allow the search space $\mathcal{X}$ to contain categorical and numerical dimensions alike and consider only sequential evaluations. We refer to a specific HPO problem with the 3-tuple $(\mathcal{X},\, f_{\mathcal{A}},\, b)$. For a discussion on potential extensions of our framework to different HPO formulations, we refer the reader to Section 6.

**Developer adjustments**   We now put developer adjustments on concrete terms and introduce a taxonomy of developer adjustments. We consider two main categories of developer adjustments: ones that do not change the search space $\mathcal{X}$ (homogeneous adjustments) and ones that do (heterogenous adjustments). Homogeneous adjustments could either change the algorithm's implementation or the hardware that the algorithm is run on. Heterogeneous adjustments can be further categorized into adjustments that add or remove a hyperparameter (hyperparameter adjustments) and adjustments that change the search space for a specific hyperparameter (range adjustments). Figure 1 shows an illustration of the adjustment types.

**Knowledge transfer across adjustments**   In general, a continuous stream of developer adjustments could be accompanied by multiple HPOs. We simplify the problem in this fundamental work and only consider the transfer between two HPO problems; we discuss a potential extension in Section 6. The two HPO problems arise from adjustments $\Psi$ to a ML algorithm $\mathcal{A}_{\text{old}}$ and its search space $\mathcal{X}_{\text{old}}$, which lead to $\mathcal{A}_{\text{new}}, \mathcal{X}_{\text{new}} := \Psi(\mathcal{A}_{\text{old}},\, \mathcal{X}_{\text{old}})$. Specifically, the hyperparameter transfer across adjustments problem is to solve the HPO problem $(\mathcal{X}_{\text{new}},\, f_{\mathcal{A}_{\text{new}}},\, b_{\text{new}})$, given the results for $(\mathcal{X}_{\text{old}},\, f_{\mathcal{A}_{\text{old}}},\, b_{\text{old}})$. Compared to HPO from scratch, developers can choose a lower budget $b_{new}$, given evidence for a transfer algorithm achieving the same performance faster.

**Relation to hyperparameter transfer across tasks (HT-AT)**   There exists an extensive research field that studies the transfer across *tasks* for HPOs (Vanschoren, 2018). The main difference to hyperparameter transfer across *adjustments* is that HT-AT assumes that there are no changes to hardware, ML algorithm, and search space, whereas HT-AA is characterized by these changes. Another difference is that most work on HT-AT considers large amounts of meta-data (up to more than a thousand previous HPO runs and function evaluations (Wang et al., 2018; Metz et al., 2020)), whereas in the basic HT-AA problem described above, only one previous HPO run is available. Compared to HT-AT approaches, the main technical challenge for HT-AA approaches are *heterogeneous* developer adjustments, as *homogeneous* HT-AA problems, where none of the adjustments changes the search space, are syntactically equivalent to HT-AT problems with only one prior HPO run. For this homogeneous HT-AA, existing approaches for HT-AT could, in principle, be applied without modification; this includes, for example the transfer acquisition function (Wistuba et al., 2018), multi-task bayesian optimization (Swersky et al., 2013), multi-task adaptive bayesian linear regression  (Perrone et al., 2018), ranking-weighted gaussian process ensemble (Feurer et al., 2018), and difference-modelling bayesian optimisation (Shilton et al., 2017). However, the low amount of meta-data in (homogeneous) HT-AA poses additional challenges.

## 3 BASELINE ALGORITHMS FOR HT-AA

In this section we present four baselines for the specific instantiation of the hyperparameter transfer across adjustments (HT-AA) framework discussed in Section 2. We resist the temptation to introduce complex approaches alongside a new research framework and instead focus on a solid foundation. Specifically, we focus on approaches that do not use any knowledge from the new HPO for the transfer and include two baselines that are modelled after actually-practiced manual strategies. We first introduce the two basic HPO algorithm that the transfer approaches build upon, then introduce two decompositions of HPO search spaces across adjustments, and finally, we present the four baselines themselves.

### 3.1 BACKGROUND: HYPERPARAMETER OPTIMIZATION ALGORITHMS

We instantiate our transfer algorithms with two different basic hyperparameter optimization algorithms, and also evaluate against these two algorithms (Section 5). Both algorithms are based on Bayesian optimization (BO): Given observations $\mathcal{D} = \{(\boldsymbol{x}_k, f_{\mathcal{A}}(\boldsymbol{x}_k)\}_{k=1}^n$, BO fits a probabilistic model $p(f_{\mathcal{A}} \mid \mathcal{D})$ for the objective function $f_{\mathcal{A}}$; then, to decide which hyperparameter setting to evaluate next, BO maximizes an acquisition function $\alpha(\boldsymbol{x} \mid \mathcal{D})$, which uses this probabilistic model. For both approaches we follow standard practice and sample configurations randomly in a fraction of cases (here 1/3). For a recent review of BO we refer to Shahriari et al. (2016). In the following, we provide details for the two basic HPO algorithms which we use.

**Tree-Structured Parzen Estimator**   One of the two algorithms that we use is the Tree-Structured Parzen Estimator (TPE) (Bergstra et al., 2011), which is the default algorithm in the popular HyperOpt package (Bergstra et al., 2013). TPE uses kernel density estimators to model the densities $l(\boldsymbol{x})$ and $g(\boldsymbol{x})$, for the probability of a given hyperparameter configuration $\boldsymbol{x}$ being worse ($l$), or better ($g$), than the best already evaluated configuration. To decide which configuration to evaluate, TPE then maximizes the acquisition function $\boldsymbol{x}^* \in \arg \max_{\boldsymbol{x} \in \mathcal{X}} g(\boldsymbol{x})/b(\boldsymbol{x})$ approximately. In our experiments, we use the TPE implementation and hyperparameter settings from Falkner et al. (2018), which requires $2(\dim(\mathcal{X}_{\text{new}}) + 1)$ evaluations to build a model.

**Bayesian optimization with Gaussian processes**   Bayesian optimization with Gaussian processes uses Gaussian process regression (Rasmussen & Williams, 2006) to fit a probabilistic model $p(f_{\mathcal{A}} \mid \mathcal{D})$. To generate a sample, we optimize the widely used expected improvement (Jones et al., 1998) acquisition function. If there are less than $\dim(\mathcal{X})$ evaluations, we sample randomly. In our experiments, we use the implementation from the SMAC3 package (Lindauer et al., 2017) with default kernels, hyperparameters, and optimization procedures. In the following we will refer to BO with Gaussian processes with GP.

### 3.2 PRELIMINARIES: SEARCH SPACE DECOMPOSITIONS

**Hyperparameter adjustments**   For hyperparameter adjustments the new search space $\mathcal{X}_{\text{new}}$ and the old search space $\mathcal{X}_{\text{old}}$ only differ in hyperparameters, not in hyperparameter ranges, so we can decompose the search spaces as $\mathcal{X}_{\text{new}} = \mathcal{X}_{\text{only-new}} \times \mathcal{X}_{\text{both}}$ and $\mathcal{X}_{\text{old}} = \mathcal{X}_{\text{both}} \times \mathcal{X}_{\text{only-old}}$, where $\mathcal{X}_{\text{both}}$ is the part of the search space that remains unchanged across adjustments (see Figure 2 for reference). All baselines use this decomposition and project the hyperparameter settings that were evaluated in the old HPO from $\mathcal{X}_{\text{old}}$ to $\mathcal{X}_{\text{both}}$.

**Range adjustments**   A range adjustment can remove values from the hyperparameter range or add values. For an adjustment of hyperparameter range $\mathcal{X}_{old}^{H_i}$ to $\mathcal{X}_{\text{new}}^{H_i}$ this can be expressed as $\mathcal{X}_{\text{new}}^{H_i} = \mathcal{X}_{\text{both}}^{H_i} \cup \mathcal{X}_{\text{both,range-only-new}}^{H_i}$ with $\mathcal{X}_{\text{both}}^{H_i} = \mathcal{X}_{old}^{H_i} \setminus \mathcal{X}_{\text{both,range-only-old}}^{H_i}$.

### 3.3 BASELINE ALGORITHMS

**Only Optimize New Hyperparameters**   A common developer strategy to manually transfer across adjustments is to set previous hyperparameters in $\mathcal{X}_{\text{both}}$ to the best setting of the previous HPO and only optimize hyperparameters in $\mathcal{X}_{\text{only-new}}$ (Agostinelli et al., 2014; Huang et al., 2017; Wu & He, 2018).

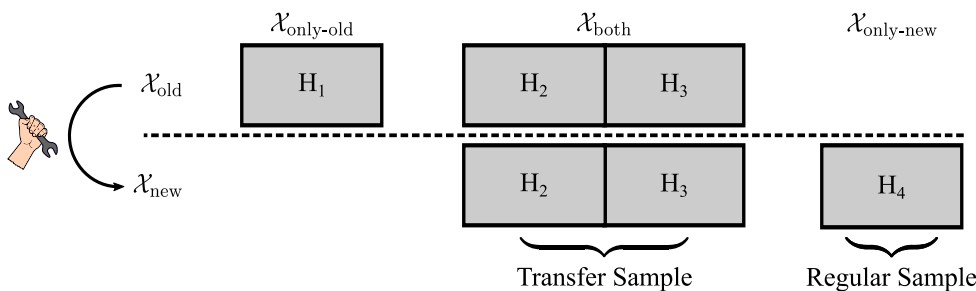

Figure 2: Example search space decomposition for a hyperparameter addition and removal.

If the previous best setting is not valid anymore, i.e., it has values in $\mathcal{X}^{H_i}_{\text{both,range-only-old}}$ for a hyperparameter $H_i$ still in $\mathcal{X}_{\text{both}}$, this strategy uses the best setting that still is a valid configuration. In the following, we refer to this strategy as *only-optimize-new*.

**Drop Unimportant Hyperparameters**   A strategy inspired by manual HT-AA efforts is to only optimize important hyperparameters. The utilization of importance statistics was, for example, explicitly disclosed in the seminal work on AlphaGo (Chen et al., 2018). Here, we determine the importance of each individual hyperparameter with functional analysis of variance (fANOVA) (Hutter et al., 2014) and do not tune hyperparameters with below mean importance. Therefore, this strategy only optimizes hyperparameters in $\mathcal{X}_{\text{only-new}}$ and hyperparameters in $\mathcal{X}_{\text{both}}$ with above mean importance. In the following, we refer to this strategy as *drop-unimportant*.

**First Evaluate Best**   The *best-first* strategy uses only-optimize-new for the first evaluation, and uses standard GP for the remaining evaluations. This strategy has a large potential speedup and low risk as it falls back to standard GP.

**Transfer GP (TGP) / Transfer TPE (T2PE)**   The main idea of Transfer GP (TGP) / Transfer TPE (T2PE) is to fit a GP / TPE model on the observations of the previous HPO $\mathcal{D}_{\text{old}} = \{(\boldsymbol{x}_k, f_{\mathcal{A}_{\text{old}}}(\boldsymbol{x}_k)\}^{b_{\text{old}}}_{k=1}$ for the part of the search space that remained unchanged (i.e., $\mathcal{X}_{\text{both}}$, for reference see Figure 2). To this end, TGP/T2PE first discards now invalid hyperparameter settings and projects the remaining settings $\boldsymbol{x}_i$ in $\mathcal{D}_{\text{old}}$ from $\mathcal{X}_{\text{old}}$ to $\mathcal{X}_{\text{both}}$ to yield $\mathcal{D}_{\text{old,both}}$. Then, either a GP or TPE model $M_{\text{both}}$ is fit on these transformed previous results $\mathcal{D}_{\text{old,both}}$, and to generate a sample $\boldsymbol{x} \in \mathcal{X}_{\text{new}}$ for the new HPO, a sample from $M_{\text{both}}$ over $\mathcal{X}_{\text{both}}$ is combined with a random sample over $\mathcal{X}_{\text{only-new}}$. Further, as in the basic HPO algorithms, TGP/T2PE uses a random sample over $\mathcal{X}_{\text{new}}$ in $^1/_3$ of cases, and after $\dim(\mathcal{X}_{\text{new}})$ (TGP) or $2(\dim(\mathcal{X}_{\text{new}}) + 1)$ (T2PE) evaluations, TGP/T2PE uses a model fitted on the new observations. We provide pseudocode for TGP/T2PE in Algorithm 1.

---

**Algorithm 1** Sampling strategy in transfer GP/TPE

---

    **Input**: Current search space $\mathcal{X}_{\text{new}}$, previous search space $\mathcal{X}_{\text{old}}$, previous results $\mathcal{D}_{\text{old}}$, budget $b_{\text{new}}$

1: Decompose $\mathcal{X}_{\text{new}} = (\mathcal{X}_{\text{both}} \cup \mathcal{X}_{\text{both,range-only-new}}) \times \mathcal{X}_{\text{only-new}}$         ▷ See Section 3.2
2: Discard hyperparameter settings in $\mathcal{D}_{\text{old}}$ that have values in $\mathcal{X}_{\text{both,range-only-new}}$
3: Project configs in $\mathcal{D}_{\text{old}}$ to space $\mathcal{X}_{\text{both}}$, to yield $\mathcal{D}_{\text{old,both}}$
4: Fit TPE or GP model $M_{\text{both}}$ for $\mathcal{X}_{\text{both}}$ on $\mathcal{D}_{\text{old,both}}$
5: **for** $t$ **in** $1, \ldots, b_{\text{new}}$ **do**
6:     **if** is random fraction **then**                              ▷ See Section 3.1
7:         Sample $\mathbf{x}_{\text{new}}$ from prior on $\mathcal{X}_{\text{new}}$
8:     **else if** no model for $\mathcal{X}_{\text{new}}$ **then**        ▷ Differs for TPE and GP, see Section 3.1
9:         Sample $\mathbf{x}_{\text{both}}$ from $\mathcal{X}_{\text{both}}$ using $M_{\text{both}}$
10:        Sample $\mathbf{x}_{\text{only-new}}$ from prior on $\mathcal{X}_{\text{only-new}}$
11:        Combine $\mathbf{x}_{\text{both}}$ with $\mathbf{x}_{\text{only-new}}$ to yield sample $\mathbf{x}_{\text{new}}$
12:     **else**
13:        Fit TPE or GP model $M_{\text{new}}$ for $\mathcal{X}_{\text{new}}$ on current observations
14:        Sample $\mathbf{x}_{\text{new}}$ from $\mathcal{X}_{\text{new}}$ using $M_{\text{new}}$

---

## 4 BENCHMARKS FOR HT-AA

We introduce eight novel benchmarks for the basic hyperparameter transfer across adjustments (HT-AA) problem discussed in Section 2. As is common in hyperparameter optimization research, we employ tabular and surrogate benchmarks to allow cheap and reproducible benchmarking (Perrone et al., 2018; Falkner et al., 2018). Tabular benchmarks achieve this with a lookup table for all possible hyperparameter settings. In contrast, surrogate benchmarks fit a surrogate model for the objective function (Eggensperger et al., 2014). We base our surrogate models and lookup tables on code and data from existing hyperparameter optimization (HPO) benchmarks (detailed in Appendix A) that, together, cover four different machine learning algorithms: a fully connected neural network (FCN), neural architecture search for a convolutional neural network (NAS), a support vector machine (SVM), and XGBoost (XGB). While the NAS and FCN benchmarks are based on lookup tables, the SVM and XGB based benchmarks use surrogate models. For each of these base benchmarks, we simulate two different types of developer adjustments (Table 1), to arrive at a total of eight benchmarks. Additionally, for each algorithm and adjustment, we consider multiple tasks in our benchmarks. Further, we provide a python package with all our benchmarks and refer the reader to Appendix A for additional details on the benchmarks.

Table 1: Developer adjustments and types of adjustments in the benchmarks.

| Benchmark | Adjustments | Adjustment Type |
|---|---|---|
| FCN-A | Increase #units-per-layer 16× | Homogeneous |
| | Double #epochs | Homogeneous |
| | Fix batch size hyperparameter | Heterogeneous |
| FCN-B | Add per-layer choice of activation function | Heterogeneous |
| | Change learning rate schedule | Homogeneous |
| NAS-A | Add 3x3 average pooling as choice of operation to each edge | Heterogeneous |
| NAS-B | Add node to cell template (adds 3 hyperparameters) | Heterogeneous |
| XGB-A | Expose four booster hyperparameters | Heterogeneous |
| XGB-B | Change four unexposed booster hyperparameter values | Homogeneous |
| SVM-A | Change kernel | Homogeneous |
| | Remove hyperparameter for old kernel | Heterogeneous |
| | Add hyperparameter for new kernel | Heterogeneous |
| SVM-B | Increase range for cost hyperparameter | Heterogeneous |

## 5 EXPERIMENTS AND RESULTS

In this section, we empirically evaluate the four baseline algorithms presented in Section 3 as solutions for the hyperparameter transfer across adjustments problem. We first describe the evaluation protocol used through all studies and then present the results.

**Evaluation protocol**   We use the benchmarks introduced in Section 4 and focus on the speedup of transfer strategies over either Bayesian optimization with Gaussian processes (GP), or over Tree Parzen Estimator (TPE). Specifically, we measured how much faster a transfer algorithm reaches a given objective value compared to GP or TPE in terms of the number of evaluations. We repeated all measurements across 25 different random seeds and report results for validation objectives, as not all benchmarks provide test objectives, and to reduce noise in our evaluation. We terminate runs after 400 evaluations and report ratio of means. To aggregate these ratios across tasks and benchmarks, we use the geometric mean. We use the geometric mean, as, intuitively, two speedups of e.g., 0.1x and 10x average to 1x, and not 5.05x. We want to note that the standard mean is an upper bound for the geometric mean, so using the geometric mean in fact makes the speedups slightly less impressive than had we used the standard arithmetic mean. To determine the target objective values, we measured GP's average performance for 10, 20, and 40 evaluations. We chose this range of evaluations as a

Table 2: Average speedup across benchmarks for different #evaluations for the old and new HPO. For the GP based (left) and TPE based (right) evaluation.

| #Evals Old | #Evals New | Best First | Transfer GP/TPE | Best First + Transfer GP/TPE |
|---|---|---|---|---|
| 10 | 10 | 1.6 / 1.5 | 1.3 / 1.0 | **1.7** / **1.7** |
|  | 20 | **1.4** / **1.4** | 1.2 / 1.0 | **1.4** / **1.4** |
|  | 40 | 1.1 / **1.2** | 1.1 / 1.1 | **1.2** / **1.2** |
| 20 | 10 | 2.3 / 1.8 | 1.7 / 1.3 | **2.6** / **2.1** |
|  | 20 | 1.7 / 1.4 | 1.5 / 1.2 | **2.0** / **1.8** |
|  | 40 | 1.3 / 1.1 | 1.2 / 1.1 | **1.4** / **1.5** |
| 40 | 10 | 3.3 / **2.6** | 2.1 / 1.5 | **3.6** / **2.6** |
|  | 20 | 2.6 / **2.0** | 2.0 / 1.4 | **2.9** / **2.0** |
|  | 40 | 1.8 / 1.4 | 1.5 / 1.2 | **2.1** / **1.5** |

survey among NeurIPS2019 and ICLR2020 authors indicates that most hyperparameter optimizations (HPOs) do not consider more than 50 evaluations (Bouthillier & Varoquaux, 2020). Further, for transfer approaches, we perform this experiment for different evaluation budgets for the HPO before the adjustments (also for 10, 20, and 40 evaluations).

**Results**    The transfer GP (TGP) / transfer TPE (T2PE) and best-first strategy lead to large speedups, while drop-unimportant and only-optimize-new perform poorly. Here, in this paragraph we focus on the GP based evaluation. On average and depending on the budgets for the old and new HPO, TGP reaches the given objective values 1.1–2.1x faster than GP, and best-first 1.1–3.3x faster. The combination of TGP/T2PE and best-first leads to further speedups over best-first if the budget for the old HPO was 20 or 40, and when the old budget and new budget both equal 10. These additional average speedups when combining TGP with best-first are between 0.1 and 0.3. We visualize these results with violin plots (Figure 3), as they take into account the multi modality of the results, and also provide a table with exact averages (Table 2). There are two main trends visible: (1) The more optimal the target objective, the smaller the speedup, and (2) the higher the budget for the previous HPO, the higher the speedup. For a more fine-grained visualization that shows violin plots over task averages for each benchmark, we refer to Appendix B. The approaches inspired by actually-practiced manual strategies, drop-unimportant and only-optimize-new, do not reach the performance of GP in a large percentage of (20%–70% on average), even while given at least 10x the budget compared to GP These high failure rates makes an evaluation for the speedup unfeasible and indicates that actually-practiced manual strategies perform worse than starting from scratch. For a visualization of the failure rates of drop-unimportant and only-optimize-new, and for GP, TGP, and best-first (0-0.8%) we refer the reader to Appendix C.

Additionally, we provide a study on the improvement in objective in Appendix D; in Appendix E we show the results of a control study that compares GP and TPE with different ranges of random seeds; and in Appendix F we compare GP and TPE to random search to validate their reliability.

## 6    RELATED WORK AND RESEARCH OPPORTUNITIES

In this section, we discuss work related to hyperparameter transfer across adjustments (HT-AA) and present several research opportunities in combining existing ideas with HT-AA.

**Transfer learning**    Transfer learning studies how to use observations from one or multiple source tasks to improve learning on one or multiple target tasks (Zhuang et al., 2019). If we view the HPO problems before and after specific developer adjustments as tasks, we can consider HT-AA as a specific transfer learning problem. As developer adjustments may change the search space, HT-AA would then be categorized as a heterogeneous transfer learning problem (Day & Khoshgoftaar, 2017).

**Transfer learning across adjustments**    Recently, Berner et al. (2019) transferred knowledge between deep reinforcement learning agents across developer adjustments. They crafted techniques to preserve, or approximately preserve, the neural network policy for each type of adjustment they

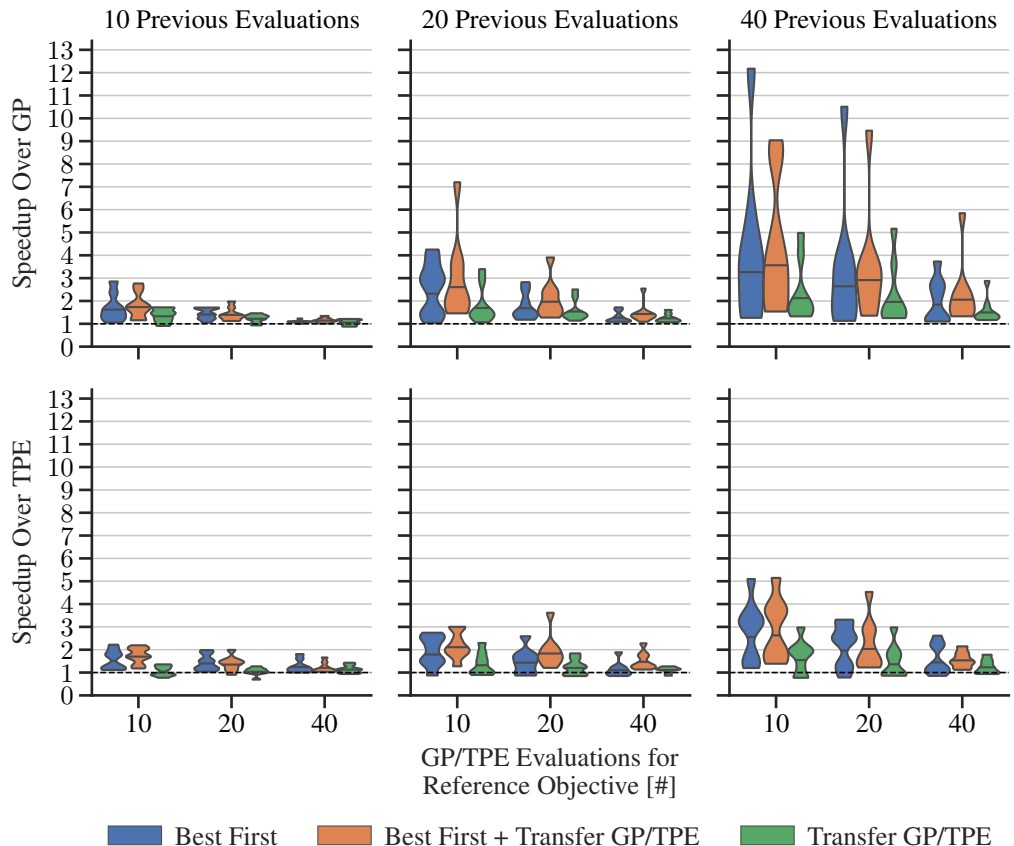

Figure 3: Speedup to reach a given reference objective value compared to GP/TPE for best-first, best-first combined with transfer GP/TPE, and transfer GP/TPE (from left to right in each violin triplet) across 8 benchmarks. The violins estimate densities of benchmark geometric means. The horizontal line in each violin shows the geometric mean across these benchmark means. #Evaluations for the old HPO increases from left to right. The x-axis shows the budget for the GP and TPE reference. Note that the GP and TPE runs we used to determine the reference objective are different to the ones that we show the speedups over.

encountered. Their transfer strategies are inspired by Net2Net knowledge transfer (Chen et al., 2015), and they use the term surgery to refer to this practice. Their work indicates that transfer learning across adjustments is not limited to knowledge about hyperparameters, but extends to a more general setting, leaving room for many research opportunities.

**Continuous knowledge transfer**    In this paper, we focus on transferring knowledge from the last HPO performed, but future work could investigate a continuous transfer of knowledge across many cycles of adjustments and HPOs. Transferring knowledge from HPO runs on multiple previous versions could lead to further performance gains, as information from each version could be useful for the current HPO. Such *continuous* HT-AA would then be related to the field of continual learning (Thrun & Mitchell, 1995; Lange et al., 2020).

**Hyperparameter transfer across tasks (HT-AT)**    While we have related HT-AA to HT-AT in Section 2, here, we want to complement this discussion with the research opportunities we see in HT-AA based on HT-AT. First, an adaptation of across-task strategies, e.g., the ones cites in Section 2, to the across-adjustments setting could lead to more powerful HT-AA approaches in the future. Second, the combination of across-task and across-adjustments hyperparameter transfer could provide even larger speedups than either transfer strategy on its own.

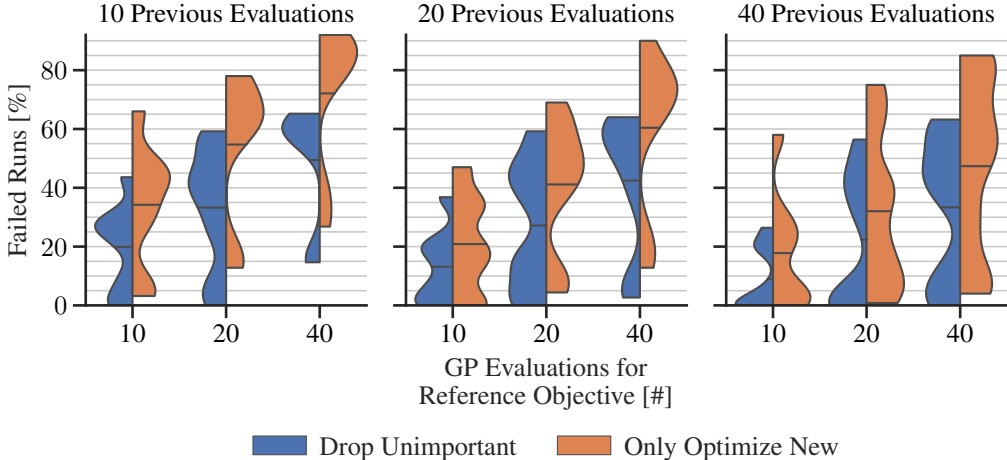

Figure 4: Percent of runs that do not reach the reference objective for the GP based drop-unimporant and only-optimize-new. Each data point for the violins represents the mean percentage of failures for a benchmark. The line in each violin shows the mean across these benchmark means. #Evaluations for the old HPO increases from left to right. The x-axis shows the budget for the GP reference.

**Advanced hyperparameter optimization**    HT-AA can be combined with the many extensions to hyperparameter optimization (HPO). One such extension is multi-fidelity HPO, which allows the use of cheap-to-evaluate approximations to the actual objective (Li et al., 2017; Falkner et al., 2018). Similarly, cost-aware HPO adds a cost to each hyperparameter setting, so the evaluation of cheap settings can be prioritized over expensive ones (Snoek et al., 2012). Yet another extension is to take evaluation noise into account (Kersting et al., 2007), or to consider not one, but multiple objectives to optimize for (Khan et al., 2002). All these HPO formulations can be studied in conjunction with HT-AA, to either provide further speedups or deal with more general optimization problems.

**Guided machine learning**    The field of guided machine learning (gML) studies the design of interfaces that enables humans to guide ML processes (Westphal et al., 2019). An HT-AA algorithm could be viewed as a ML algorithm that receives incremental guidance in the form of arbitrary developer adjustments; the interface would then be the programming language(s) the ML algorithm is implemented in. On a different note, gML could provide HT-AA algorithms with additional information about the adjustments to the ML algorithm. For example, when adding a hyperparameter, there are two distinctions we can make: Either an existing hyperparameter is exposed , or a new component is added to the algorithm that introduces a new hyperparameter From the HPO problem itself, we cannot know which case it is, and neither which fixed value an exposed hyperparameter had. Guided HT-AA algorithms could ask for user input to fill this knowledge gap, or HT-AA algorithms with code analysis could extract this knowledge from the source code.

**Programming by optimization**    The programming by optimization (PbO) framework (Hoos, 2012) proposes the automatic construction of a search space of algorithms based on code annotations, and the subsequent search in this search space. While PbO considers changing search spaces over developer actions, each task and development step restarts the search from scratch. This is in contrast to HT-AA, which alleviates the need to restart from scratch after each developer adjustment.

## 7    CONCLUSION

In this work, we introduced hyperparameter transfer across developer adjustments to improve efficiency during the development of ML algorithms. In light of rising energy demands of ML algorithms and rising global temperatures, more efficient ML development practices are an important issue now and will become more important in the future. As already some of the simple baseline algorithm considered in this work lead to large empirical speedups, our new framework represents a promising step towards efficient ML development.

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

# A    BENCHMARK SUITE DETAILS

## A.1    OVERVIEW

Table 3: Benchmarks overview

| Benchmark | #Hyperparameters Old | #Hyperparameters New | #Tasks | Metric |
|---|---|---|---|---|
| FCN-A | 6 | 5 | 4 | MSE |
| FCN-B | 6 | 8 | 4 | MSE |
| NAS-A | 6 | 6 | 3 | Accuracy |
| NAS-B | 3 | 6 | 3 | Accuracy |
| XGB-A | 5 | 9 | 10 | AUC |
| XGB-B | 6 | 6 | 10 | AUC |
| SVM-A | 2 | 2 | 10 | AUC |
| SVM-B | 2 | 2 | 10 | AUC |

## A.2    FCN-A & FCN-B

**Base benchmark**    We use code and data from (Klein & Hutter, 2019). We use the tasks HPO-Bench-Protein, HPO-Bench-Slice, HPO-Bench-Naval, and HPO-Bench-Parkinson.

**Budget**    For FCN-A the budget is set to 100. For FCN-B, additional to the changes in the search space (Table 6), the budget is increased from 50 to 100 epochs.

Table 4: Values for integer coded hyperparameters in FCN benchmarks

| Hyperparameter | Values |
|---|---|
| # Units Layer $\{1, 2\}$ | (16, 32, 64, 128, 256, 512) |
| Dropout Layer $\{1, 2\}$ | (0.0, 0.3, 0.6) |
| Initial Learning Rate | (0.0005, 0.001, 0.005, 0.01, 0.05, 0.1) |
| Batch Size | (8, 16, 32, 64) |

Table 5: Search spaces in FCN-A. Numerical hyperparameters are encoded as integers, see Table 4 for specific values for these hyperparameters.

| Steps | Hyperparameter | Range/Value | Prior |
|---|---|---|---|
| 1 | # Units Layer 1 | 1 | - |
| 1 | # Units Layer 2 | 1 | - |
| 1 | Batch Size | $\{0, \dots, 3\}$ | Uniform |
| 1, 2 | Dropout Layer 1 | $\{0, \dots, 2\}$ | Uniform |
| 1, 2 | Dropout Layer 2 | $\{0, \dots, 2\}$ | Uniform |
| 1, 2 | Activation Layer 1 | $\{ReLu, tanh\}$ | Uniform |
| 1, 2 | Activation Layer 2 | $\{ReLu, tanh\}$ | Uniform |
| 1, 2 | Initial Learning Rate | $\{0, \dots, 5\}$ | Uniform |
| 1, 2 | Learning Rate Schedule | Constant | Uniform |
| 2 | # Units Layer 1 | 5 | - |
| 2 | # Units Layer 2 | 5 | - |
| 2 | Batch Size | 1 | - |

Table 6: Search spaces in FCN-B. Numerical hyperparameters are encoded as integers, see Table 4 for specific values for these hyperparameters.

| Steps | Hyperparameter | Range/Value | Prior |
|---|---|---|---|
| 1 | Activation Layer 1 | tanh | - |
| 1 | Activation Layer 2 | tanh | - |
| 1 | Learning Rate Schedule | Constant | - |
| 1, 2 | # Units Layer 1 | $\{0, \ldots, 5\}$ | Uniform |
| 1, 2 | # Units Layer 2 | $\{0, \ldots, 5\}$ | Uniform |
| 1, 2 | Dropout Layer 1 | $\{0, \ldots, 2\}$ | Uniform |
| 1, 2 | Dropout Layer 2 | $\{0, \ldots, 2\}$ | Uniform |
| 1, 2 | Initial Learning Rate | $\{0, \ldots, 5\}$ | Uniform |
| 1, 2 | Batch Size | $\{0, \ldots, 3\}$ | Uniform |
| 2 | Activation Layer 1 | {ReLu, tanh} | Uniform |
| 2 | Activation Layer 2 | {ReLu, tanh} | Uniform |
| 2 | Learning Rate Schedule | Cosine | - |

## A.3 NAS-A & NAS-B

**Base benchmark** We use code and data from (Dong & Yang, 2019). We use the tasks CIFAR10, CIFAR100, and ImageNet.

Table 7: Search spaces in NAS-A.

| Steps | Hyperparameter | Range/Value | Prior |
|---|---|---|---|
| 1, 2 | $0 \rightarrow 2$ | { none, skip-connect, conv1x1, conv3x3, avg-pool3x3 } | Uniform |
| 1, 2 | $0 \rightarrow 3$ | { none, skip-connect, conv1x1, conv3x3, avg-pool3x3 } | Uniform |
| 1, 2 | $2 \rightarrow 3$ | { none, skip-connect, conv1x1, conv3x3, avg-pool3x3 } | Uniform |
| 2 | $0 \rightarrow 1$ | { none, skip-connect, conv1x1, conv3x3, avg-pool3x3 } | Uniform |
| 2 | $1 \rightarrow 2$ | { none, skip-connect, conv1x1, conv3x3, avg-pool3x3 } | Uniform |
| 2 | $1 \rightarrow 3$ | { none, skip-connect, conv1x1, conv3x3, avg-pool3x3 } | Uniform |

Table 8: Search spaces in NAS-B.

| Steps | Hyperparameter | Range/Value | Prior |
|---|---|---|---|
| 1 | $0 \rightarrow 1$ | { none, skip-connect, conv1x1, conv3x3 } | Uniform |
| 1 | $0 \rightarrow 2$ | { none, skip-connect, conv1x1, conv3x3 } | Uniform |
| 1 | $0 \rightarrow 3$ | { none, skip-connect, conv1x1, conv3x3 } | Uniform |
| 1 | $1 \rightarrow 2$ | { none, skip-connect, conv1x1, conv3x3 } | Uniform |
| 1 | $1 \rightarrow 3$ | { none, skip-connect, conv1x1, conv3x3 } | Uniform |
| 1 | $2 \rightarrow 3$ | { none, skip-connect, conv1x1, conv3x3 } | Uniform |
| 2 | $0 \rightarrow 1$ | { none, skip-connect, conv1x1, conv3x3, avg-pool3x3 } | Uniform |
| 2 | $0 \rightarrow 2$ | { none, skip-connect, conv1x1, conv3x3, avg-pool3x3 } | Uniform |
| 2 | $0 \rightarrow 3$ | { none, skip-connect, conv1x1, conv3x3, avg-pool3x3 } | Uniform |
| 2 | $1 \rightarrow 2$ | { none, skip-connect, conv1x1, conv3x3, avg-pool3x3 } | Uniform |
| 2 | $1 \rightarrow 3$ | { none, skip-connect, conv1x1, conv3x3, avg-pool3x3 } | Uniform |
| 2 | $2 \rightarrow 3$ | { none, skip-connect, conv1x1, conv3x3, avg-pool3x3 } | Uniform |

## A.4 SVM-A & SVM-B

**Base benchmark** We use an open source implementation by the HPOlib authors following Perrone et al. (2018). This implementation uses data from Kühn et al. (2018) and employs a random forest as a

surrogate model (Eggensperger et al., 2014). For our benchmarks, we randomly selected the ten tasks monks-problems-2, tic-tac-toe, kc2, monks-problems-1, qsar-biodeg, nomao, pc3, bank-marketing, ada-agnostic, and hill-valley.

Table 9: Search spaces in SVM-A.

| Steps | Hyperparameter | Range/Value | Prior |
|---|---|---|---|
| 1 | Kernel | Radial | - |
| 1 | Degree | $\{2, \ldots, 5\}$ | Uniform |
| 1, 2 | Cost | $[2^{-10}, 2^{10}]$ | Log-uniform |
| 2 | Kernel | Polynomial | - |
| 2 | $\gamma$ | $[2^{-5}, 2^5]$ | Log-uniform |

Table 10: Search spaces in SVM-B.

| Steps | Hyperparameter | Range/Value | Prior |
|---|---|---|---|
| 1 | Cost | $[2^{-5}, 2^5]$ | Log-uniform |
| 1, 2 | $\gamma$ | 1 | - |
| 1, 2 | Degree | 5 | - |
| 1, 2 | Kernel | $\{$Polynomial, Linear, Radial$\}$ | Uniform |
| 2 | Cost | $[2^{-10}, 2^{10}]$ | Log-uniform |

## A.5 XGB-A & XGB-B

**Base benchmark** We use an open source implementation by the HPOlib authors following Perrone et al. (2018). This implementation uses data from Kühn et al. (2018) and employs a random forest as a surrogate model (Eggensperger et al., 2014). For our benchmarks, we randomly selected the ten tasks monks-problems-2, tic-tac-toe, kc2, monks-problems-1, qsar-biodeg, nomao, pc3, bank-marketing, ada-agnostic, hill-valley

Table 11: Search spaces in XGB-A

| Steps | Hyperparameter | Range/Value | Prior |
|---|---|---|---|
| 1 | Colsample-by-tree | 1 | - |
| 1 | Colsample-by-level | 1 | - |
| 1 | Minimum child weight | 1 | - |
| 1 | Maximum depth | 6 | - |
| 1, 2 | Booster | Tree | - |
| 1, 2 | # Rounds | $\{1, \ldots, 5,000\}$ | Uniform |
| 1, 2 | Subsample | $[0, 1]$ | Uniform |
| 1, 2 | Eta | $[2^{-10}, 2^0]$ | Log-uniform |
| 1, 2 | Lambda | $[2^{-10}, 2^{10}]$ | Log-uniform |
| 1, 2 | Alpha | $[2^{-10}, 2^{10}]$ | Log-uniform |
| 2 | Colsample-by-tree | $[0, 1]$ | Uniform |
| 2 | Colsample-by-level | $[0, 1]$ | Uniform |
| 2 | Minimum child weight | $[2^0, 2^7]$ | Log-uniform |
| 2 | Maximum depth | $\{1, \ldots, 15\}$ | Uniform |

Table 12: Search spaces in XGB-B

| Steps | Hyperparameter | Range/Value | Prior |
|---|---|---|---|
| 1 | Colsample-by-tree | 1 | - |
| 1 | Colsample-by-level | 1 | - |
| 1 | Minimum child weight | 1 | - |
| 1 | Maximum depth | 6 | - |
| 1, 2 | Booster | { Linear, Tree } | - |
| 1, 2 | # Rounds | $\{1, \ldots, 5,000\}$ | Uniform |
| 1, 2 | Subsample | $[0, 1]$ | Uniform |
| 1, 2 | Eta | $[2^{-10}, 2^0]$ | Log-uniform |
| 1, 2 | Lambda | $[2^{-10}, 2^{10}]$ | Log-uniform |
| 1, 2 | Alpha | $[2^{-10}, 2^{10}]$ | Log-uniform |
| 2 | Colsample-by-tree | 1 | - |
| 2 | Colsample-by-level | 0.5 | - |
| 2 | Minimum child weight | 10 | - |
| 2 | Maximum depth | 10 | - |

# B DETAILED SPEEDUPS

## B.1 GP BASED EVALUATION

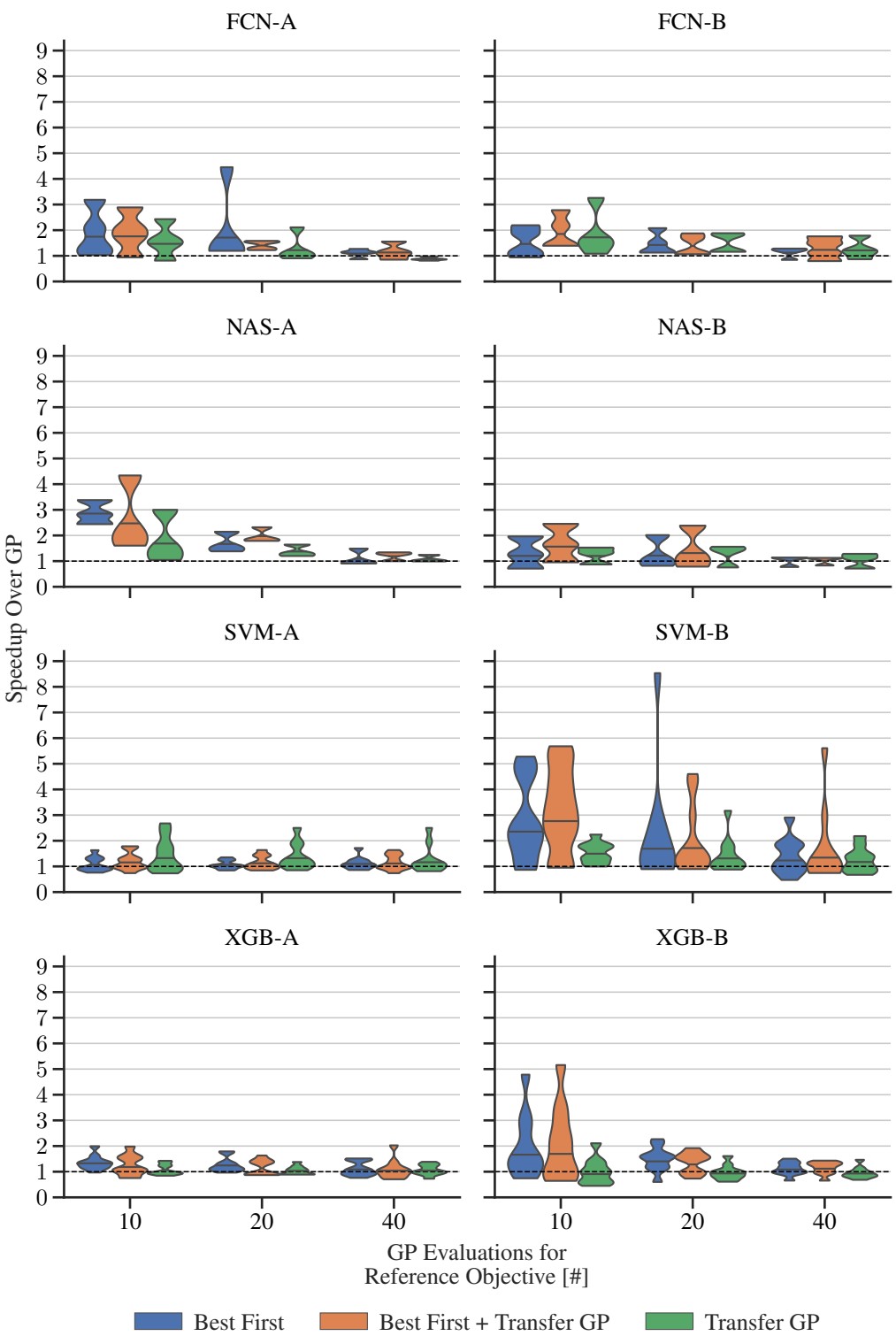

Figure 5: Speedup compared to GP for best-first, best-first combined with transfer GP, and transfer GP across tasks for each of 8 benchmarks. The previous HPO has a budget of 10 evaluations here.

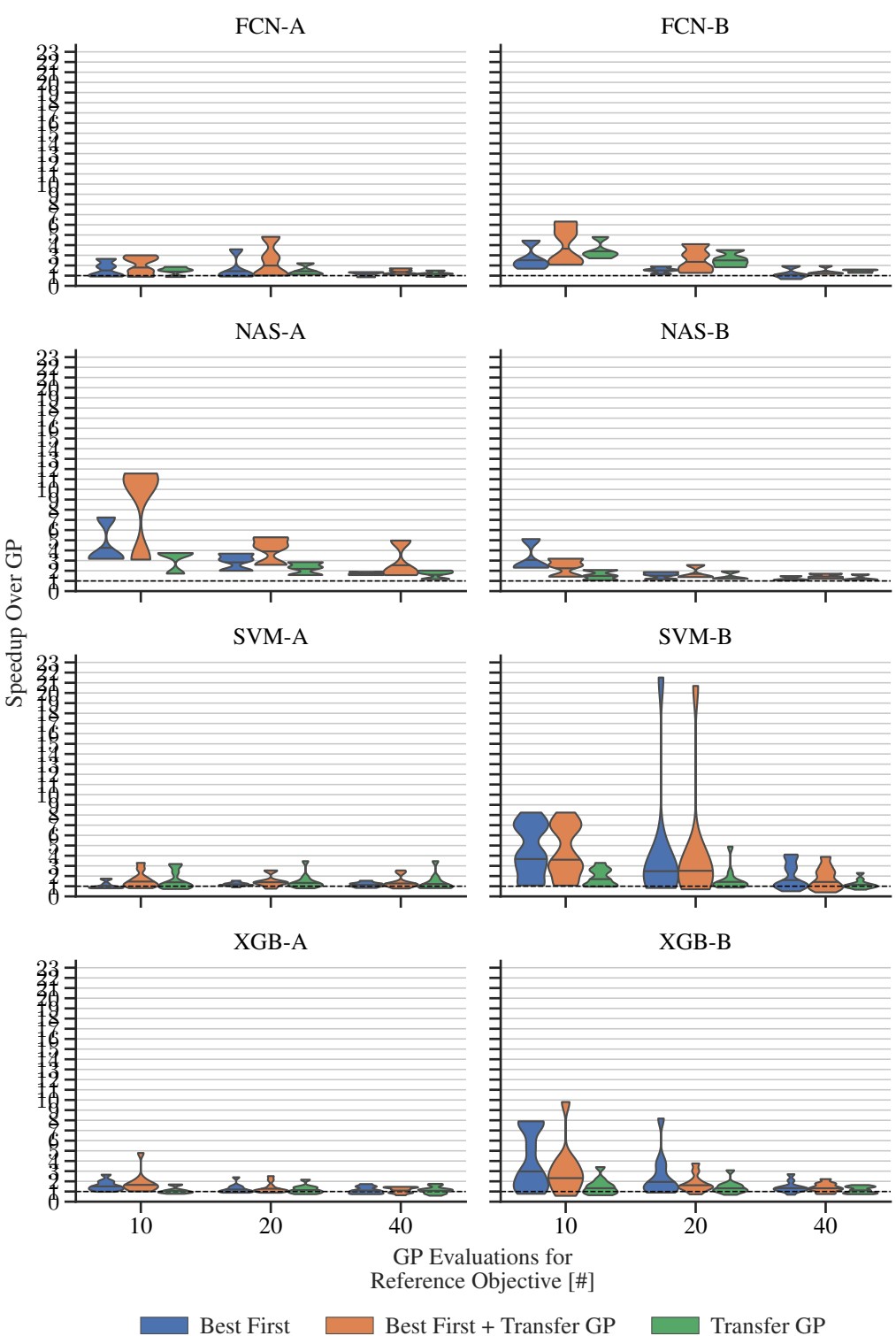

Figure 6: Speedup compared to GP for best-first, best-first combined with transfer GP, and transfer GP (from left to right in each violin triplet) across tasks for each of 8 benchmarks. The previous HPO has a budget of 20 evaluations.

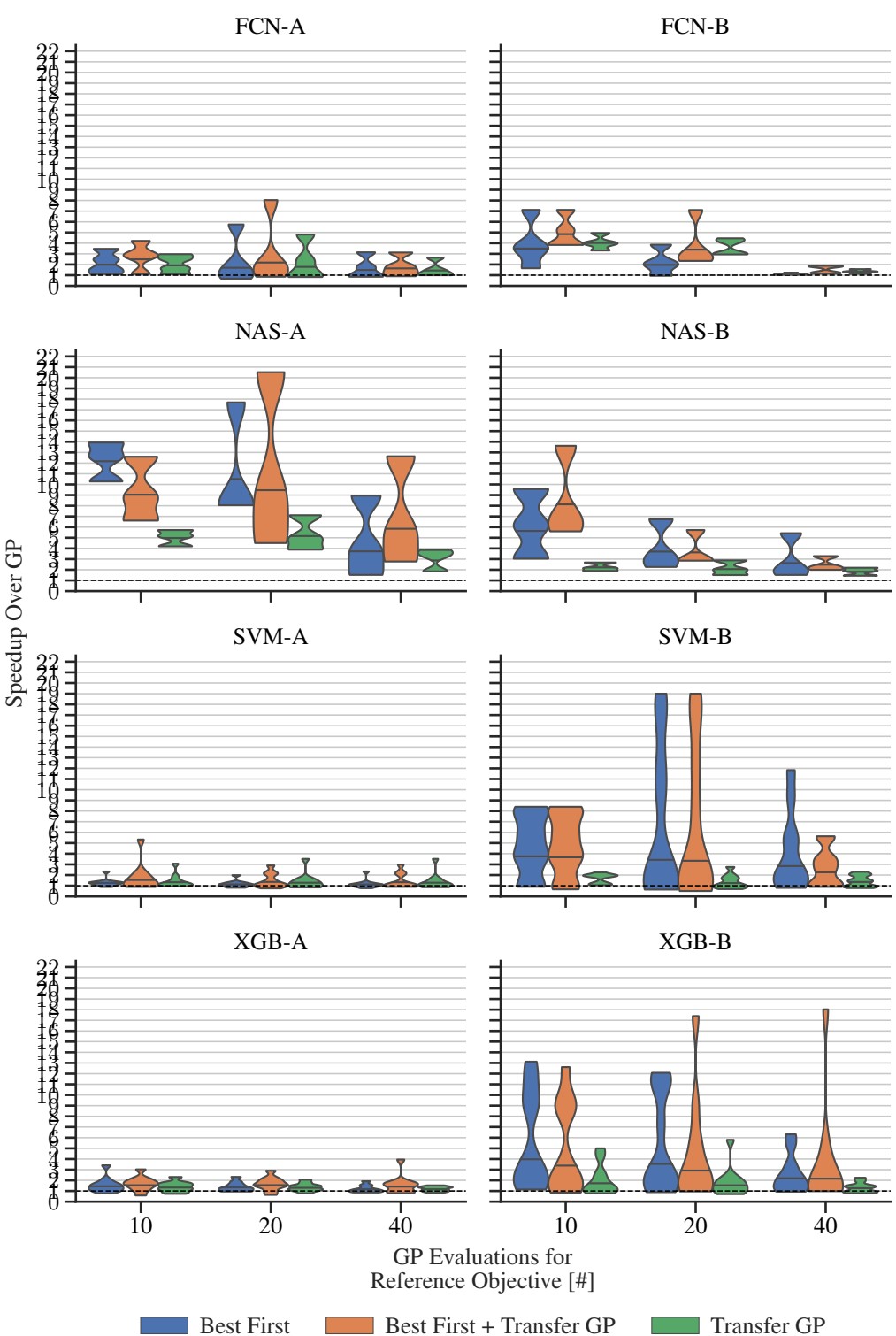

Figure 7: Speedup compared to GP for best-first, best-first combined with transfer GP, and transfer GP (from left to right in each violin triplet) across tasks for each of 8 benchmarks. The previous HPO has a budget of 40 evaluations.

## B.2 TPE BASED EVALUATION

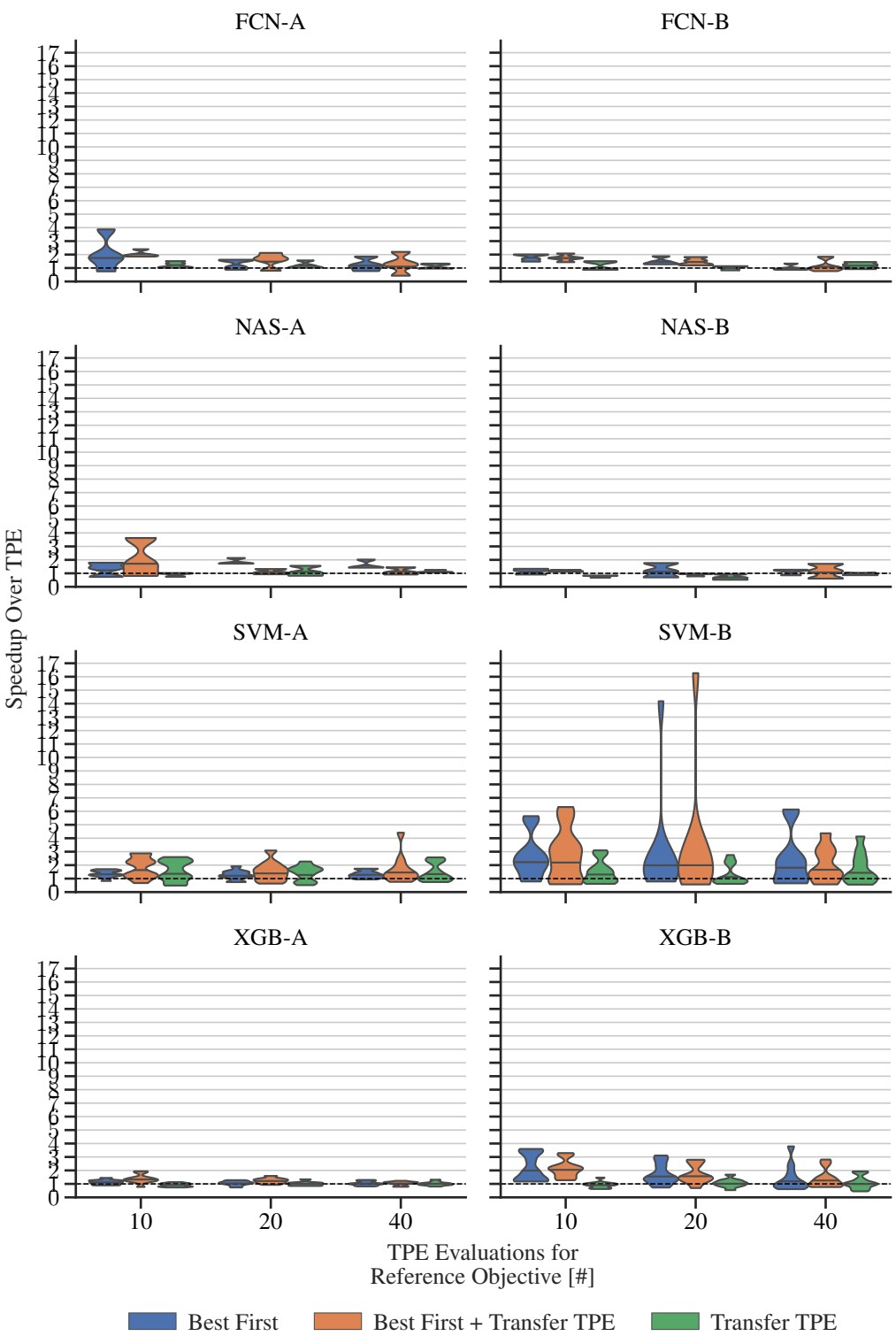

Figure 8: Speedup compared to GP for best-first, best-first combined with transfer GP, and transfer GP across tasks for each of 8 benchmarks. The previous HPO has a budget of 10 evaluations here.

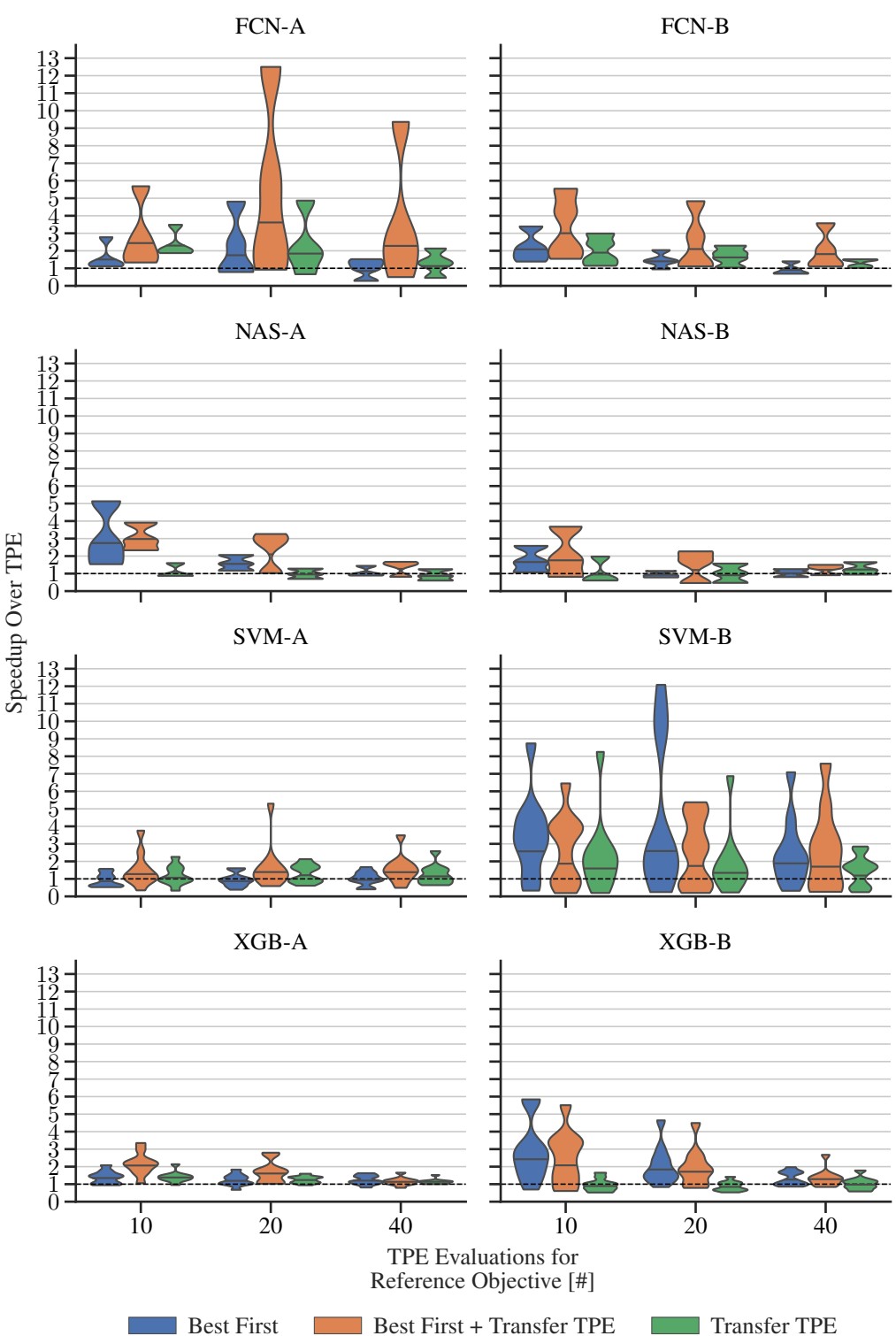

Figure 9: Speedup compared to GP for best-first, best-first combined with transfer GP, and transfer GP (from left to right in each violin triplet) across tasks for each of 8 benchmarks. The previous HPO has a budget of 20 evaluations.

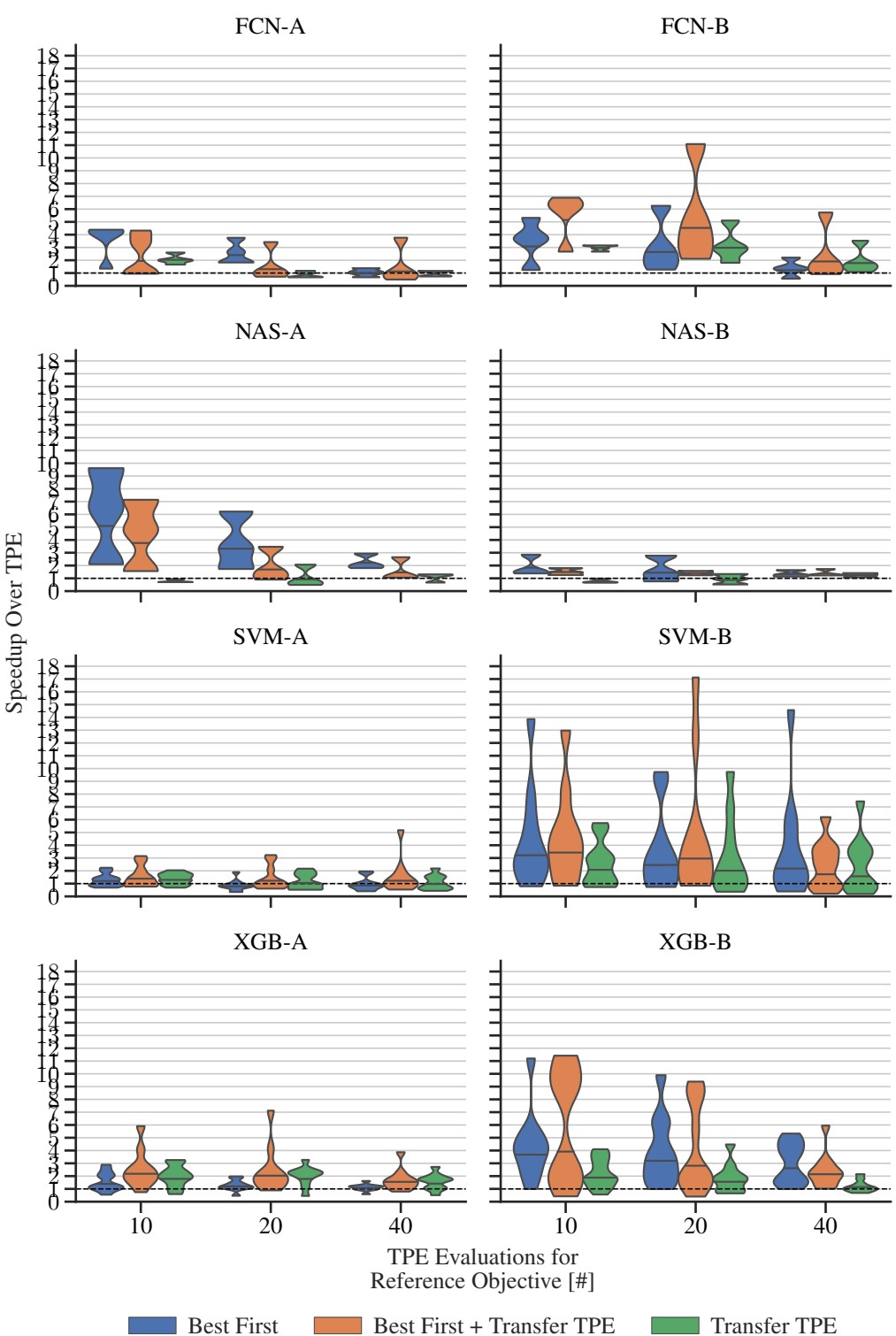

Figure 10: Speedup compared to GP for best-first, best-first combined with transfer GP, and transfer GP (from left to right in each violin triplet) across tasks for each of 8 benchmarks. The previous HPO has a budget of 40 evaluations.

## C   FAILURE RATES

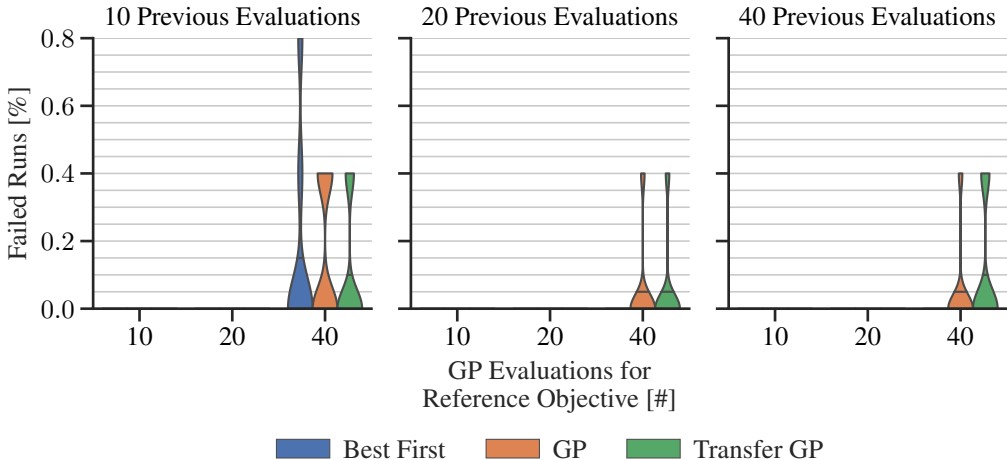

Figure 11: Failure rates for best-first, GP, and transfer GP (from left to right in each violin triplet) across 8 benchmarks. The violins estimate densities of the task means. The horizontal line in each violin shows the mean across these task means. The plots from left to right utilize increasing budget for the pre-adjustment hyperparameter. The x-axis shows the budget for the GP reference.

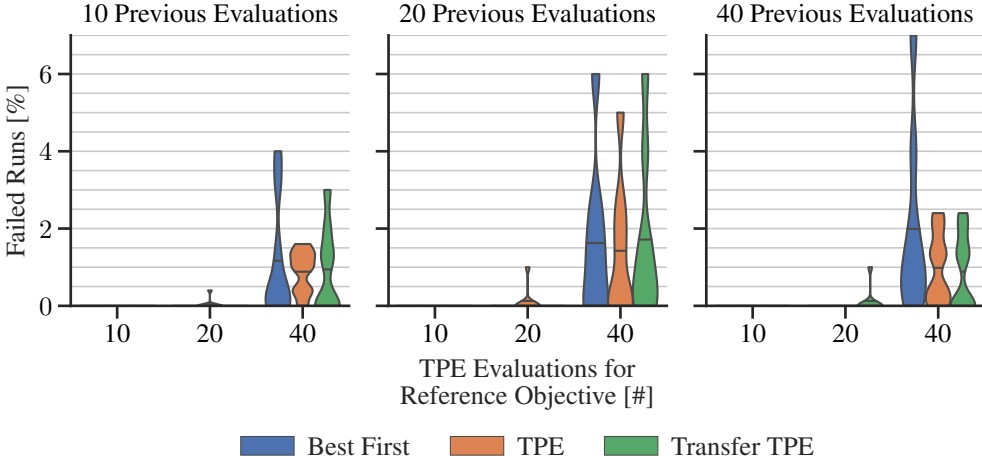

Figure 12: Failure rates for best-first, TPE, and transfer TPE (from left to right in each violin triplet) across 8 benchmarks. The violins estimate densities of the task means. The horizontal line in each violin shows the mean across these task means. The plots from left to right utilize increasing budget for the pre-adjustment hyperparameter. The x-axis shows the budget for the GP reference.

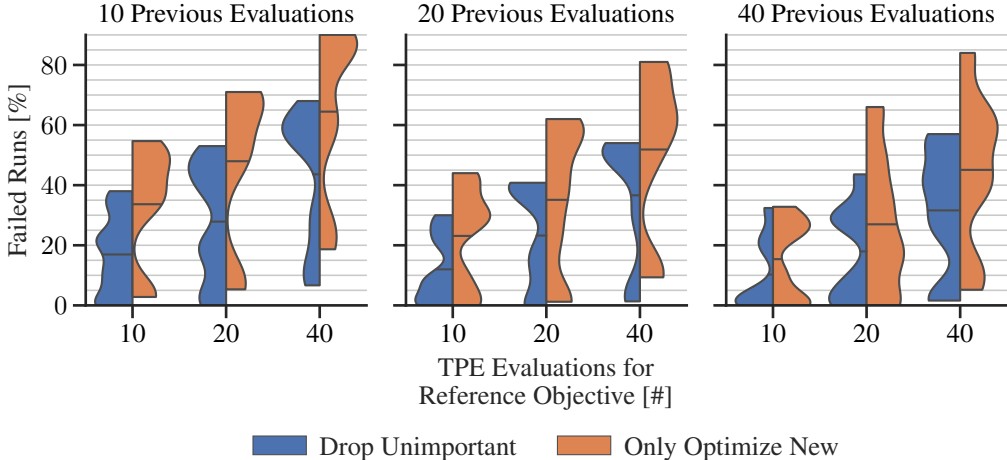

Figure 13: Failure rates for the TPE based drop-unimportant and only-optimize-new across 8 benchmarks. The violins estimate densities of the task means. The horizontal line in each violin shows the mean across these task means. The plots from left to right utilize increasing budget for the pre-adjustment hyperparameter. The x-axis shows the budget for the GP reference.

# D  OBJECTIVE IMPROVEMENTS

Here, we show the standardized improvement over the control algorithm. We compute the standardized mean improvement by averaging the final performance across repetitions, then subtract the mean performance of the control algorithm, and divide by the standard deviation of the control algorithm. This metric is known as glass' delta and intuitively gives the improvement of an algorithm over a control algorithm in the standard units of this control algorithm. As some benchmarks had a standard deviation of 0, we added a small constant in those cases. We chose this constant according to the 0.2-quantile of the observed values. For the plots we clip the improvement to $[-10, \infty)$, as for some plots there are extreme outliers. We provide results for two aggregation levels: the distribution over task standardized mean improvements, where we compute the standardized mean improvement over repetitions and show results on a per-benchmark level; and the distribution over benchmark means, i.e., the means across task standardized mean improvements for all tasks in a given benchmark.

## D.1 TRANSFER GP, BEST FIRST, AND TRANSFER GP COMBINED WITH BEST FIRST VS. GP

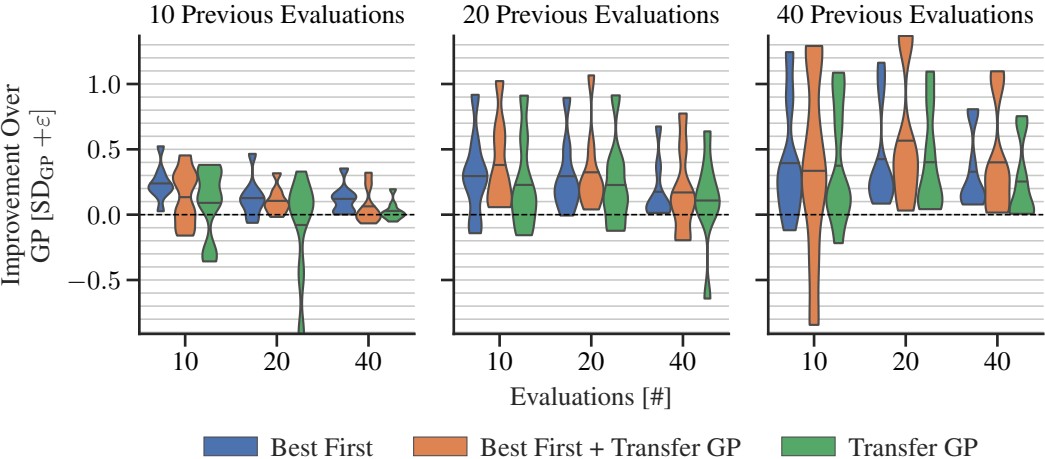

Figure 14: Standardized objective improvements of best-first, best-first combined with transfer GP, and transfer GP (from left to right in each violin triplet) over GP across 8 benchmarks. The violins estimate densities of the benchmark means. The horizontal line in each violin shows the mean across these benchmark means. #Evaluations for the old HPO increases from left to right. In each plot, the evaluation budget increases.

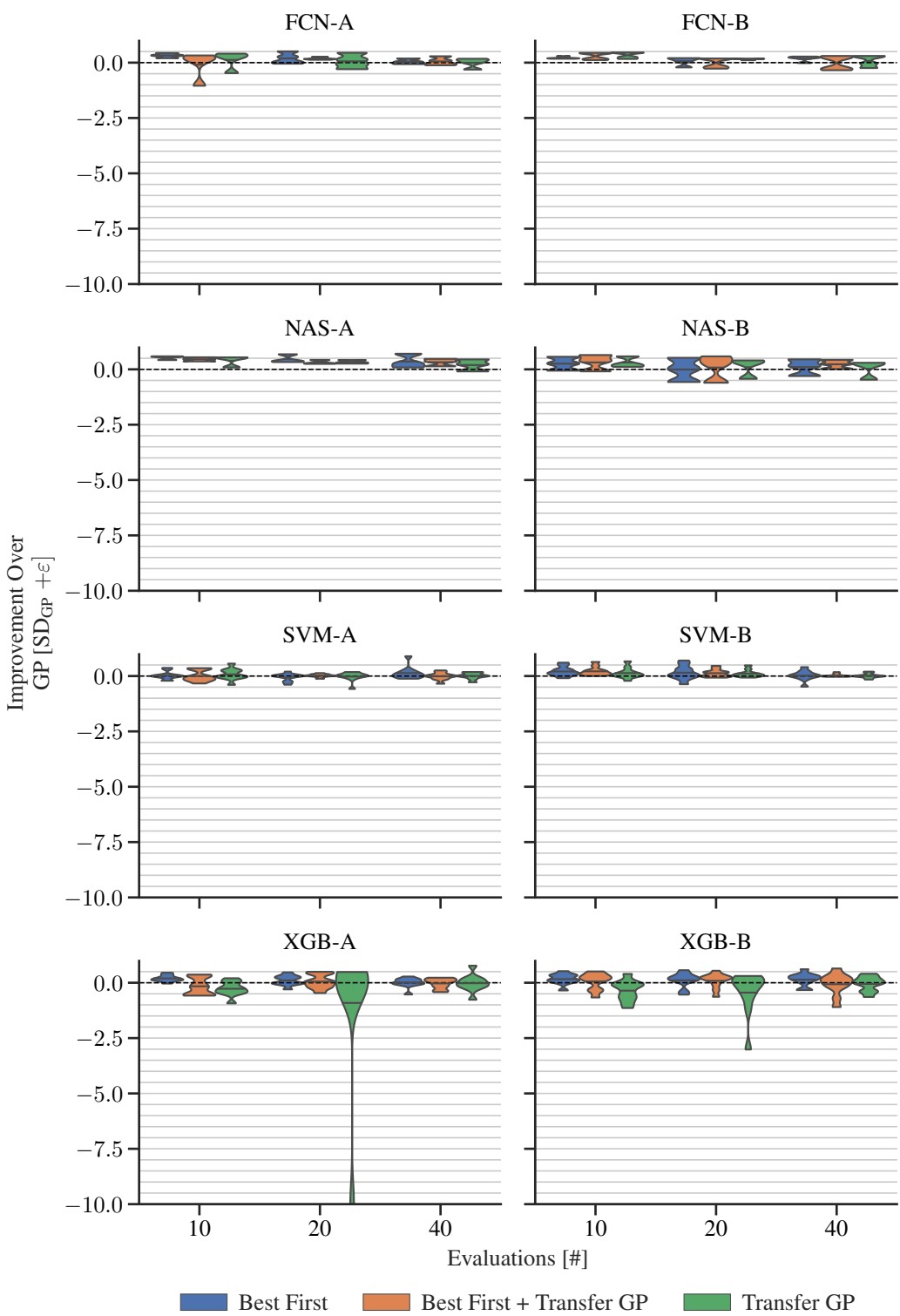

Figure 15: Standardized objective improvements of best-first, best-first combined with transfer GP, and transfer GP (from left to right in each violin triplet) across tasks for each of 8 benchmarks. The previous HPO has a budget of 10 evaluations.

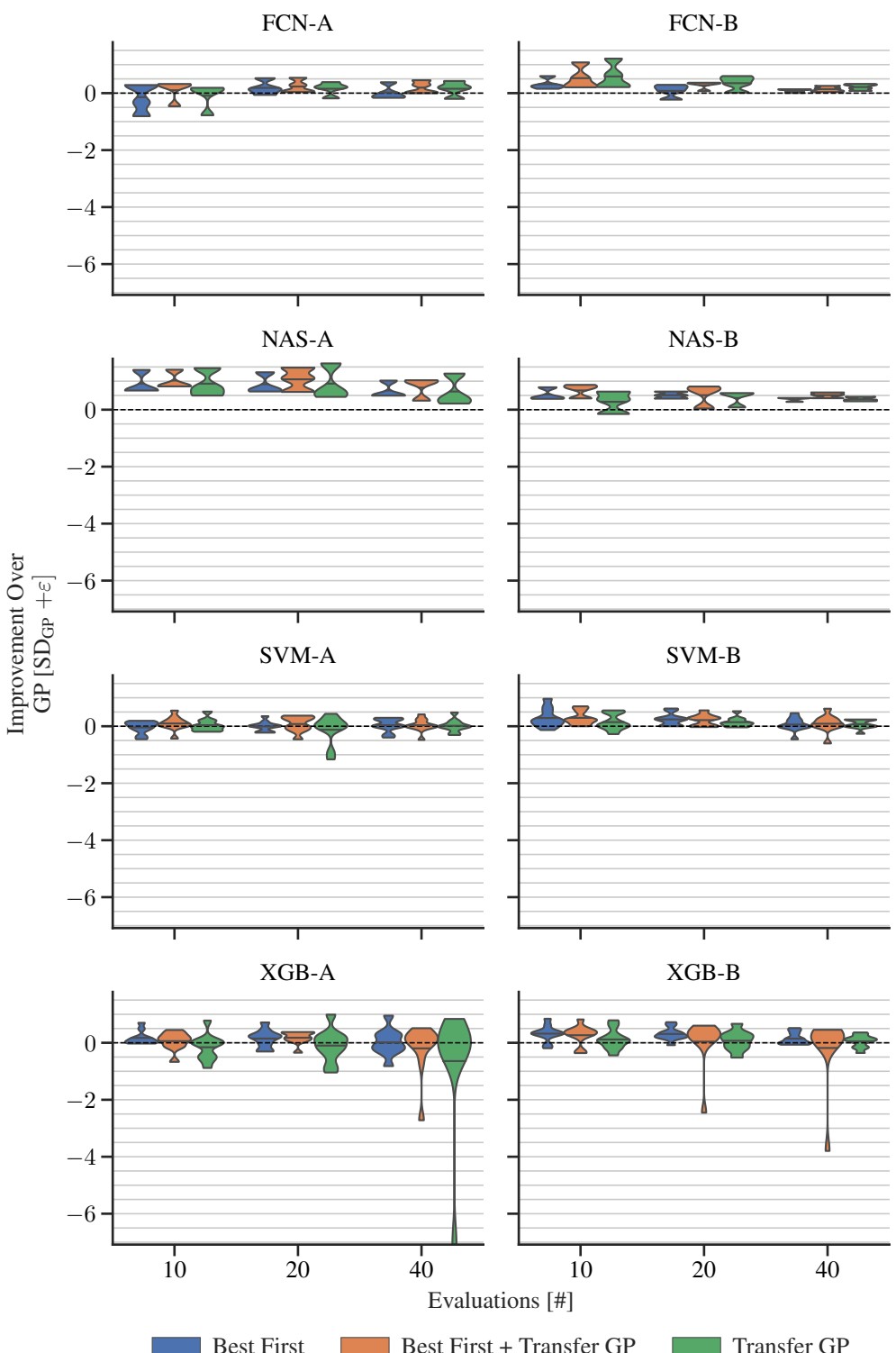

Figure 16: Standardized objective improvements of best-first, best-first combined with transfer GP, and transfer GP (from left to right in each violin triplet) across tasks for each of 8 benchmarks. The previous HPO has a budget of 20 evaluations.

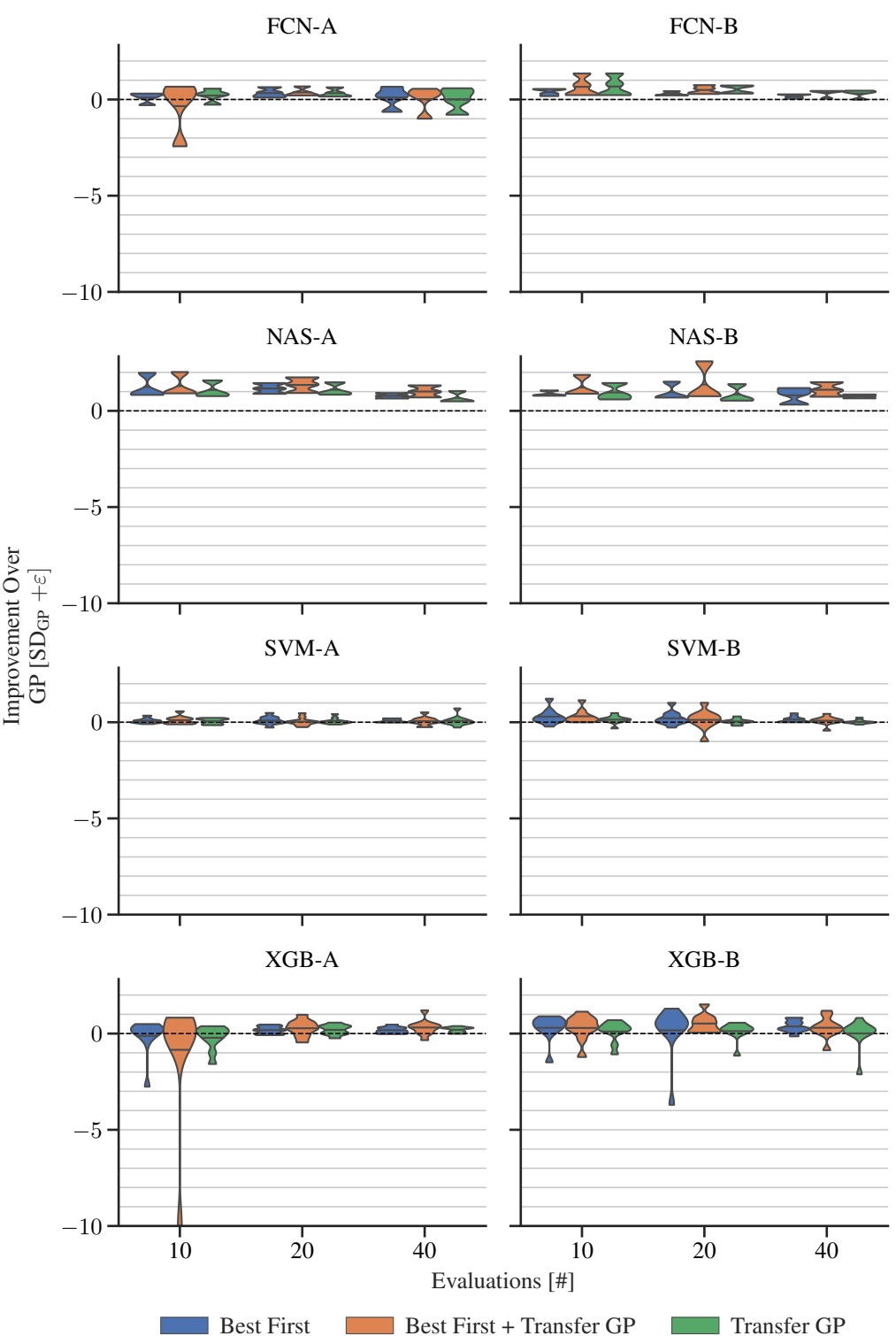

Figure 17: Standardized objective improvements of best-first, best-first combined with transfer GP, and transfer GP (from left to right in each violin triplet) across tasks for each of 8 benchmarks. The previous HPO has a budget of 40 evaluations.

## D.2 Transfer TPE, Best First, and Transfer TPE combined with Best First vs. TPE

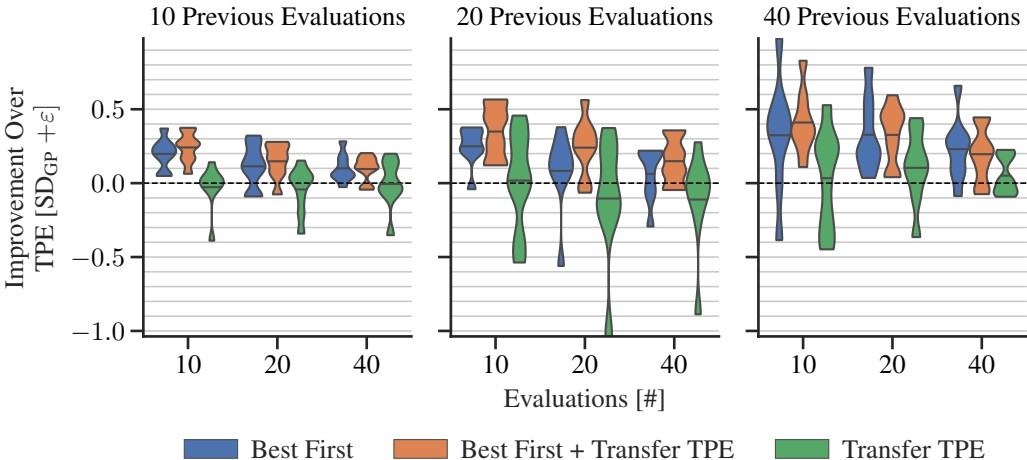

Figure 18: Standardized objective improvements of best-first, best-first combined with transfer TPE, and transfer TPE (from left to right in each violin triplet) over GP across 8 benchmarks. The violins estimate densities of the benchmark means. The horizontal line in each violin shows the mean across these benchmark means. #Evaluations for the old HPO increases from left to right. In each plot, the evaluation budget increases.

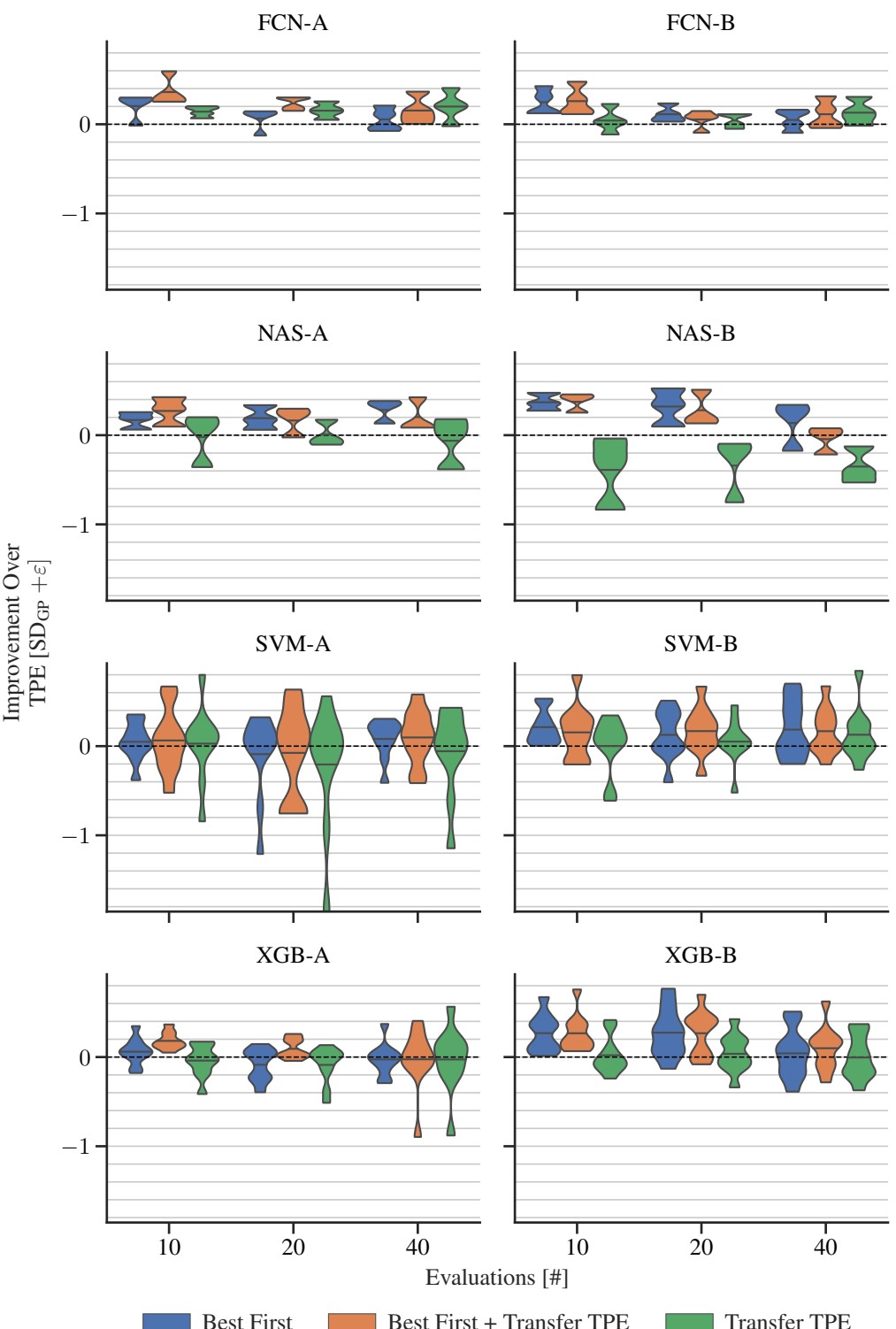

Figure 19: Standardized objective improvements of best-first, best-first combined with transfer TPE, and transfer TPE (from left to right in each violin triplet) across tasks for each of 8 benchmarks. The previous HPO has a budget of 10 evaluations.

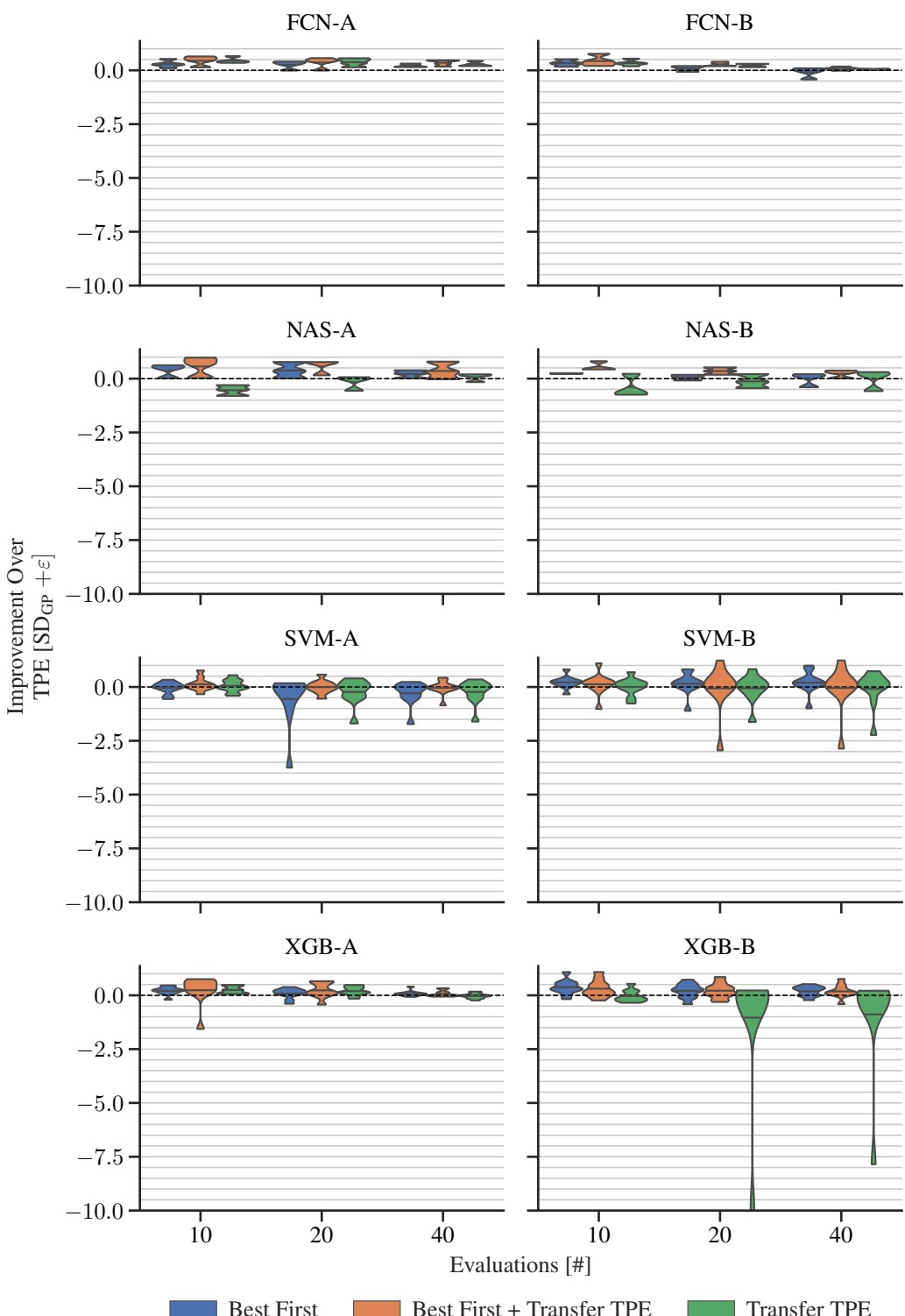

Figure 20: Standardized objective improvements of best-first, best-first combined with transfer TPE, and transfer TPE (from left to right in each violin triplet) across tasks for each of 8 benchmarks. The previous HPO has a budget of 20 evaluations.

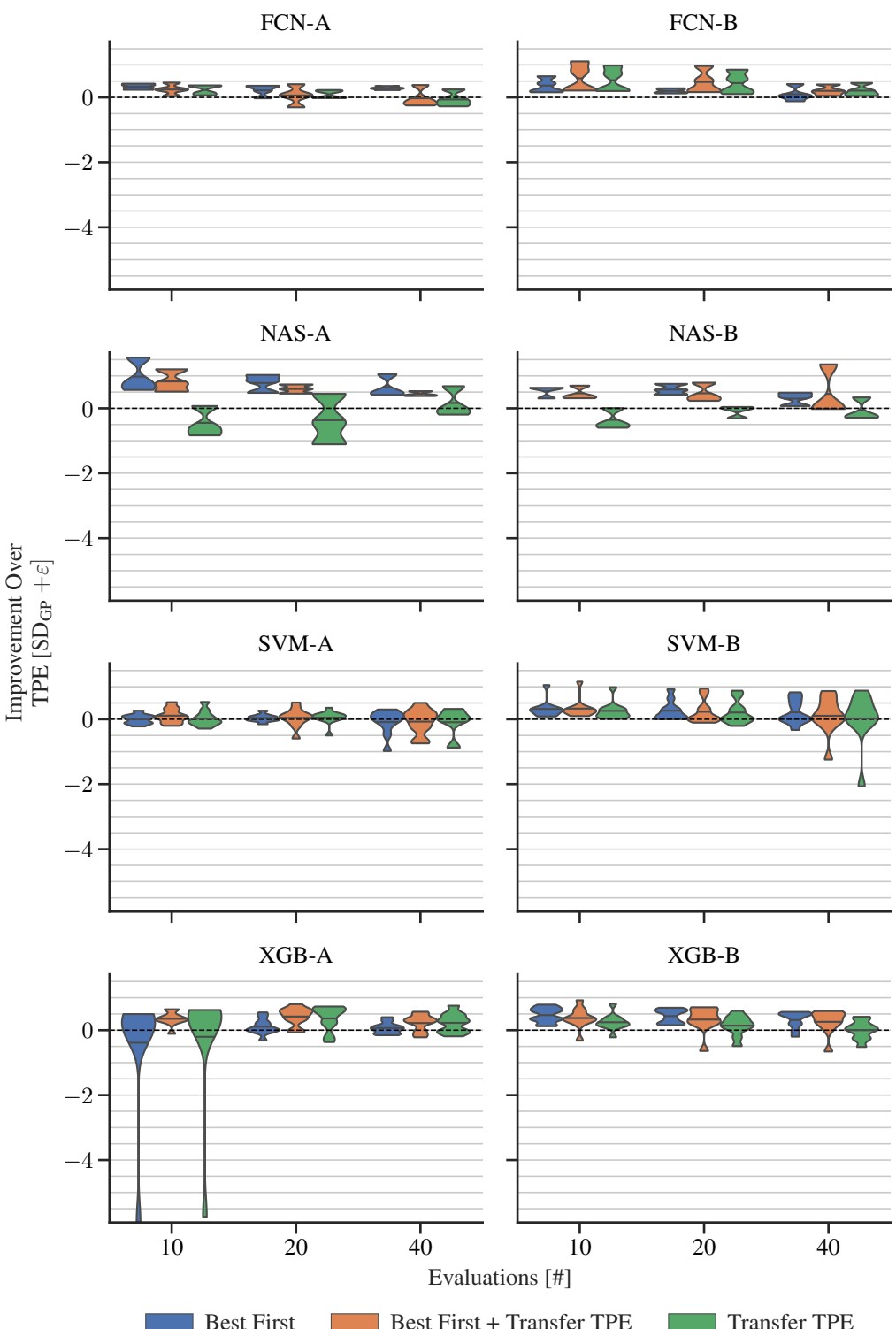

Figure 21: Standardized objective improvements of best-first, best-first combined with transfer TPE, and transfer TPE (from left to right in each violin triplet) across tasks for each of 8 benchmarks. The previous HPO has a budget of 40 evaluations.

### D.3 GP Based Only Optimize New and Drop Unimportant vs. GP

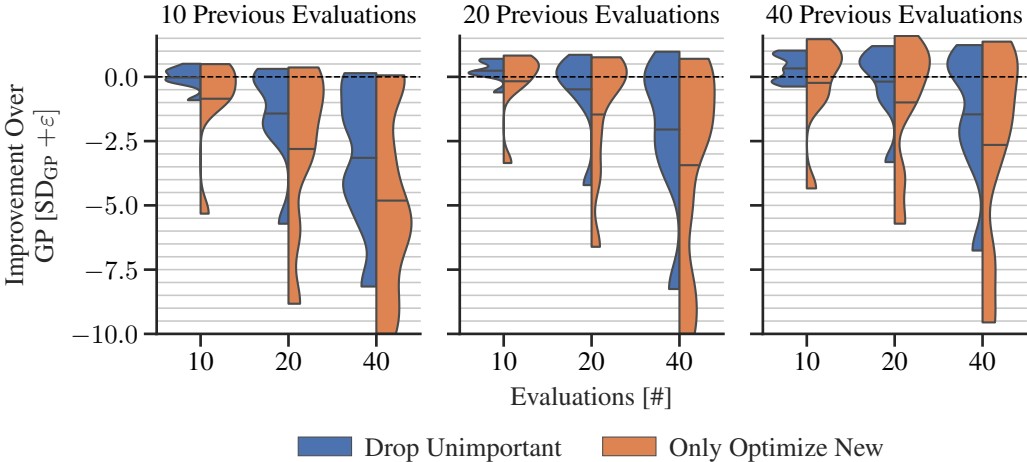

Figure 22: Standardized objective improvements of only-optimize-new and drop-unimportant over GP across 8 benchmarks. The violins estimate densities of the benchmark means. The horizontal line in each violin shows the mean across these benchmark means. #Evaluations for the old HPO increases from left to right. In each plot, the evaluation budget increases.

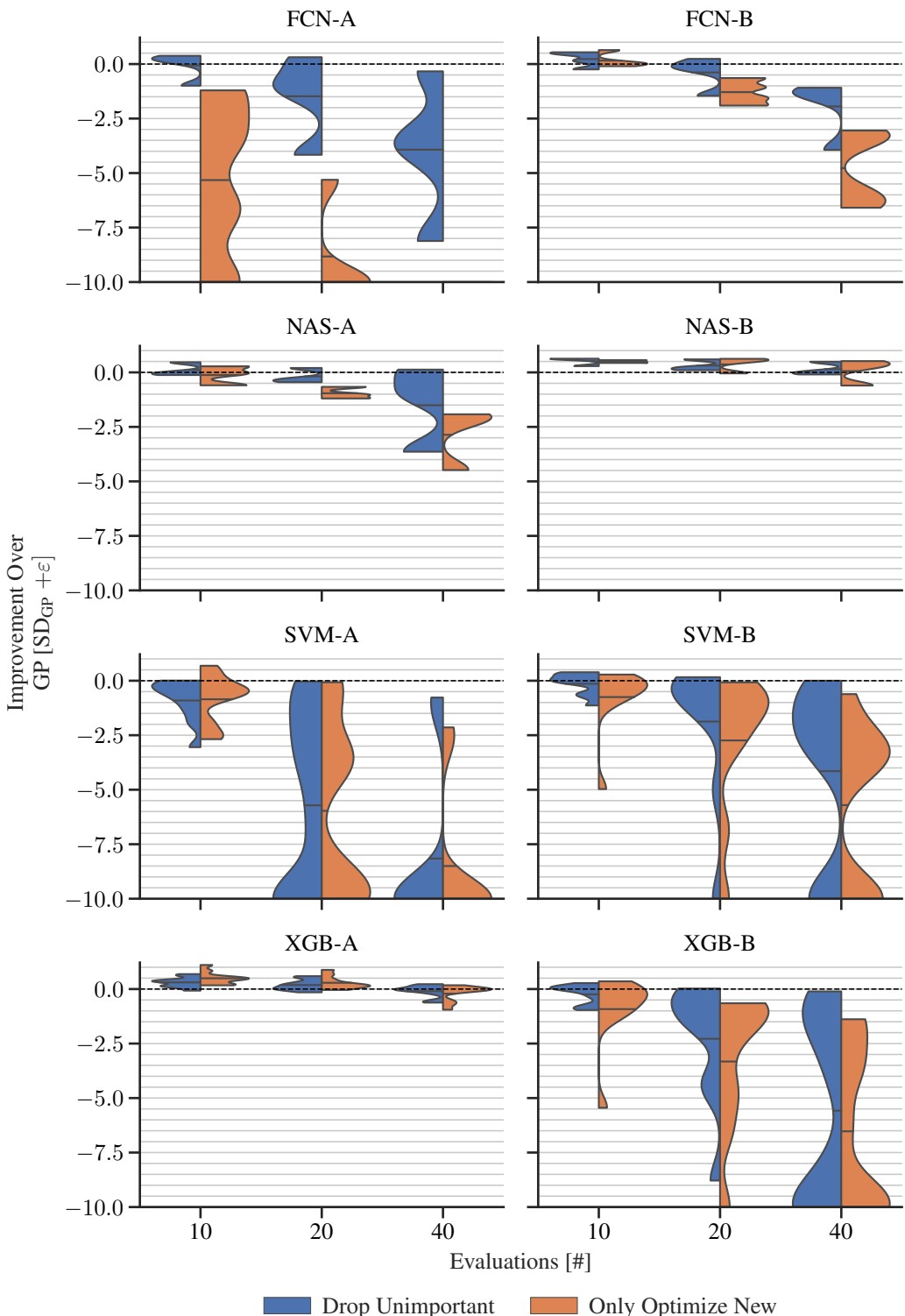

Figure 23: Standardized objective improvements of only-optimize-new and drop-unimportant across tasks for each of 8 benchmarks. The previous HPO has a budget of 10 evaluations.

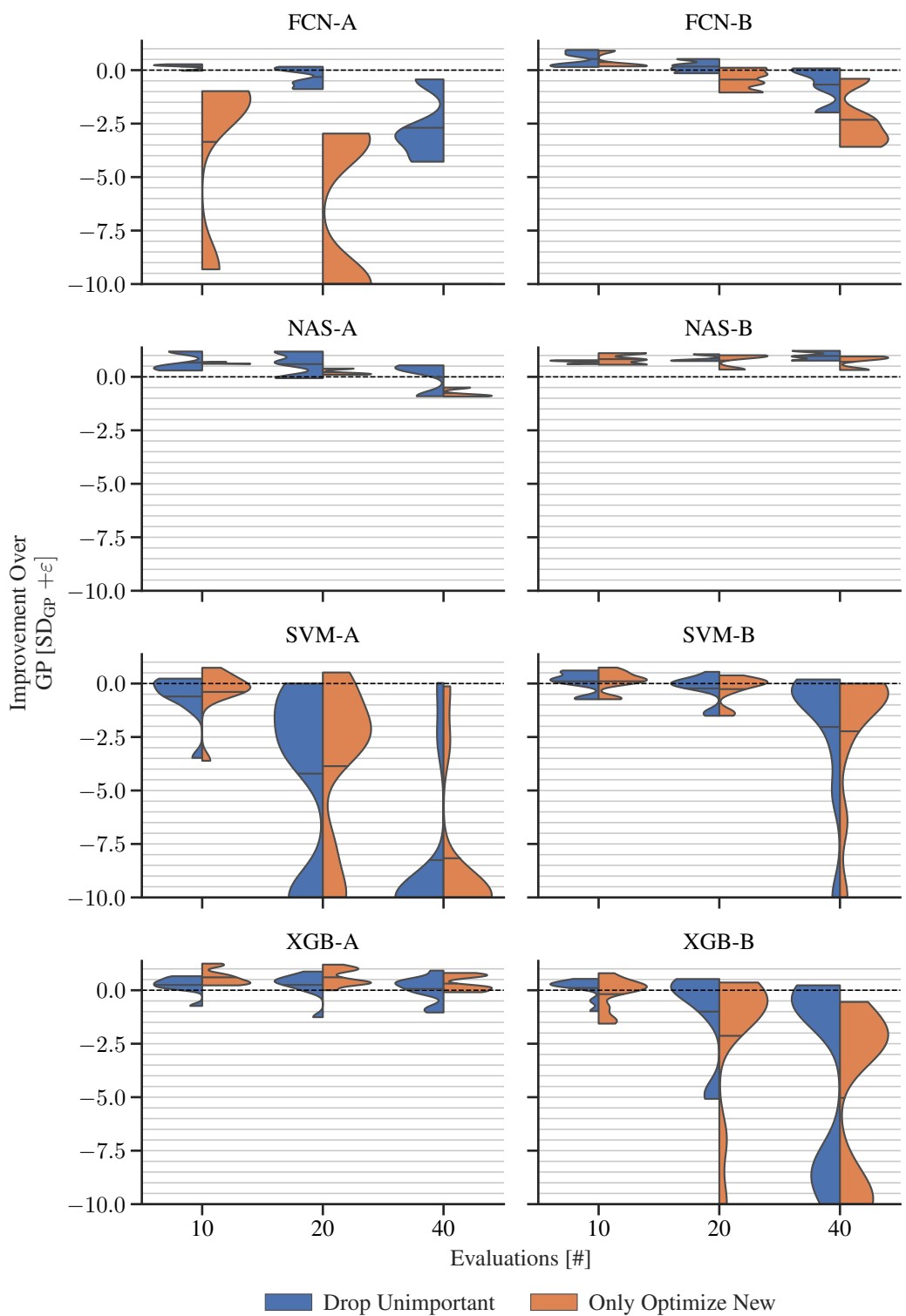

Figure 24: Standardized objective improvements of only-optimize-new and drop-unimportant across tasks for each of 8 benchmarks. The previous HPO has a budget of 20 evaluations.

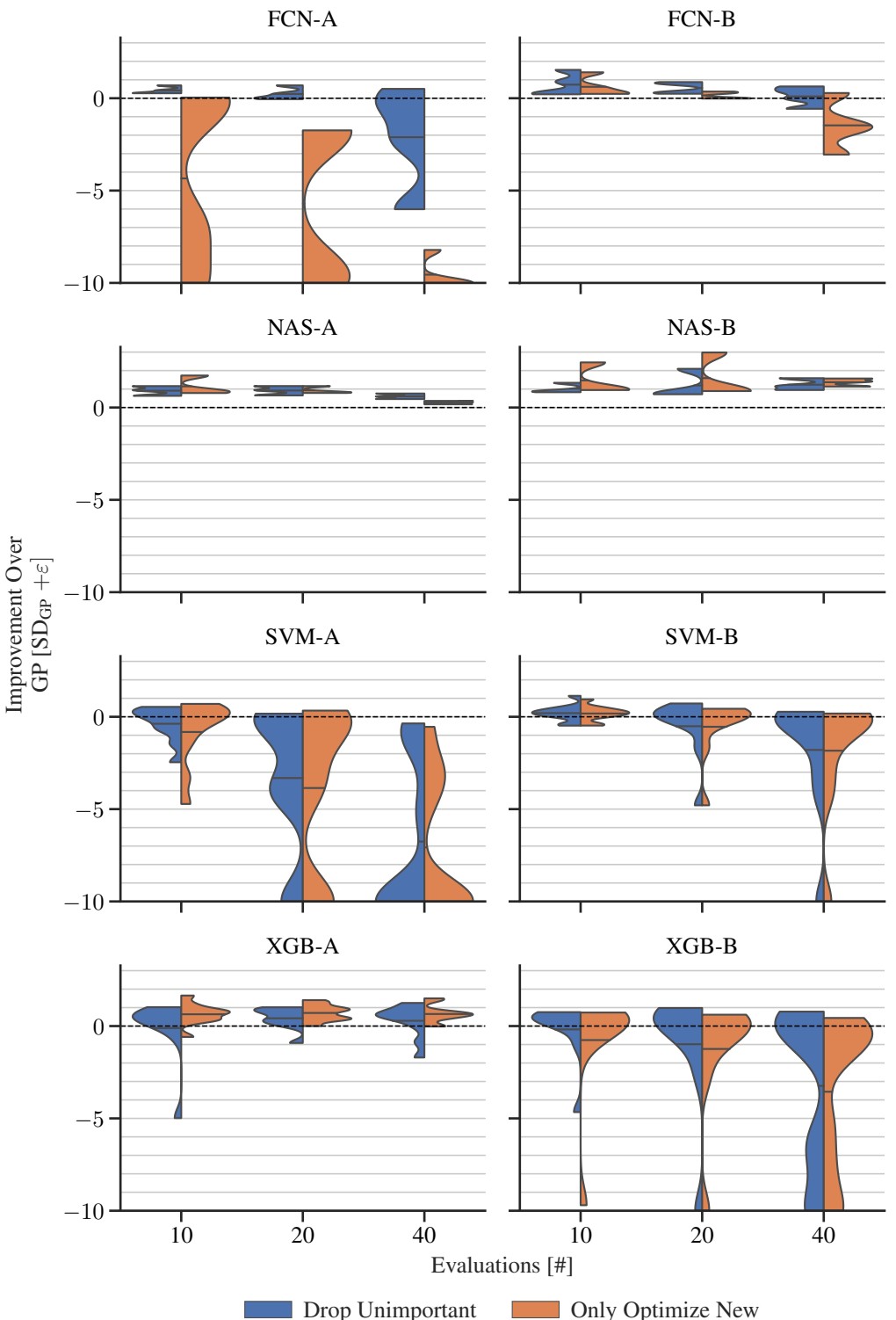

Figure 25: Standardized objective improvements of only-optimize-new and drop-unimportant across tasks for each of 8 benchmarks. The previous HPO has a budget of 40 evaluations.

### D.4 TPE BASED ONLY OPTIMIZE NEW AND DROP UNIMPORTANT VS. TPE

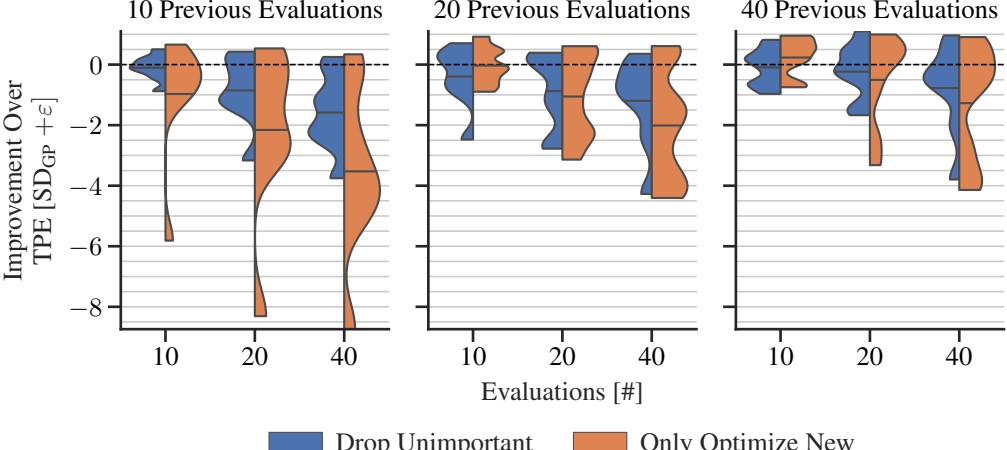

Figure 26: Standardized objective improvements of only-optimize-new and drop-unimportant over TPE across 8 benchmarks. The violins estimate densities of the benchmark means. The horizontal line in each violin shows the mean across these benchmark means. #Evaluations for the old HPO increases from left to right. In each plot, the evaluation budget increases.

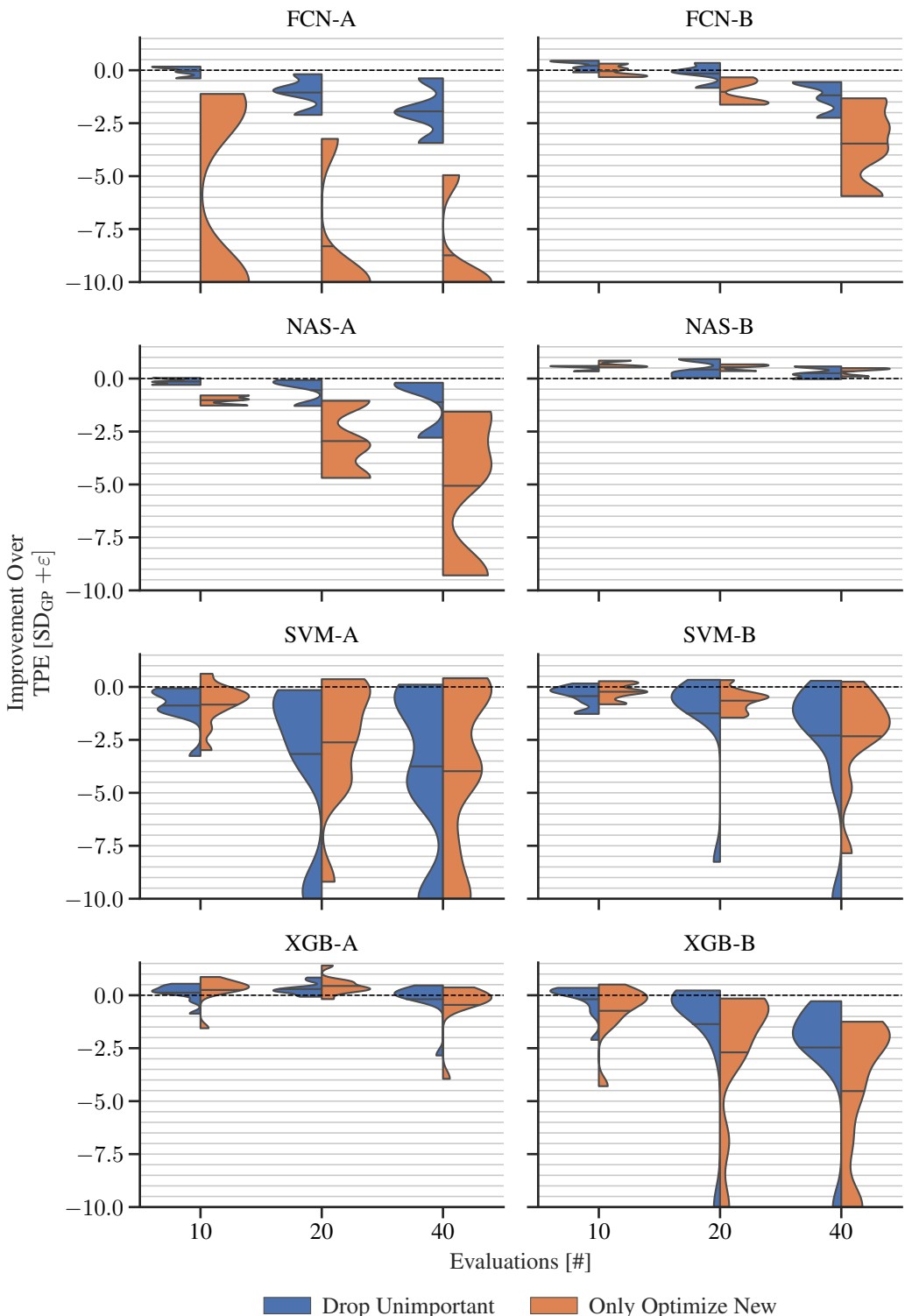

Figure 27: Standardized objective improvements of only-optimize-new and drop-unimportant across tasks for each of 8 benchmarks. The previous HPO has a budget of 10 evaluations.

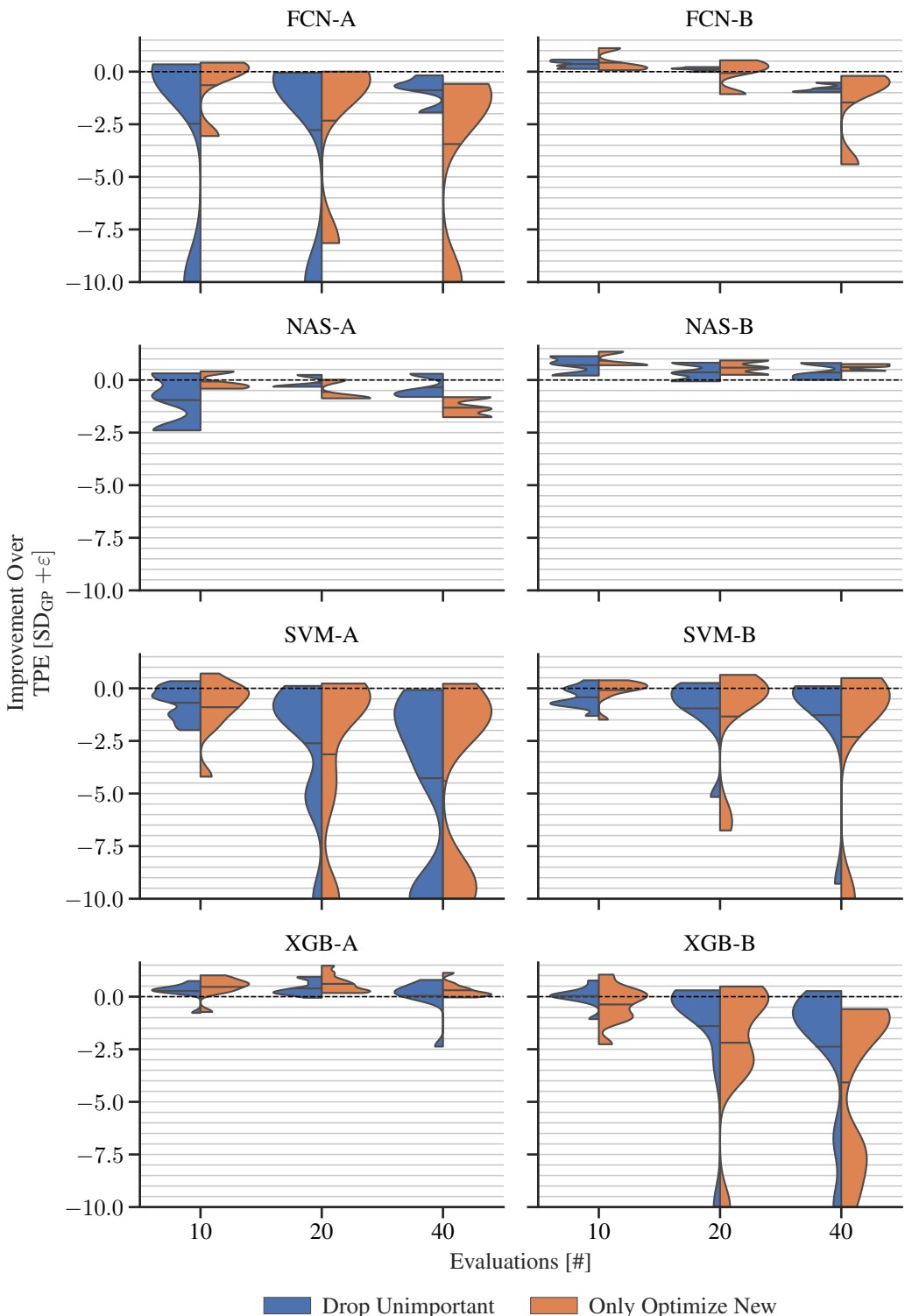

Figure 28: Standardized objective improvements of only-optimize-new and drop-unimportant across tasks for each of 8 benchmarks. The previous HPO has a budget of 20 evaluations.

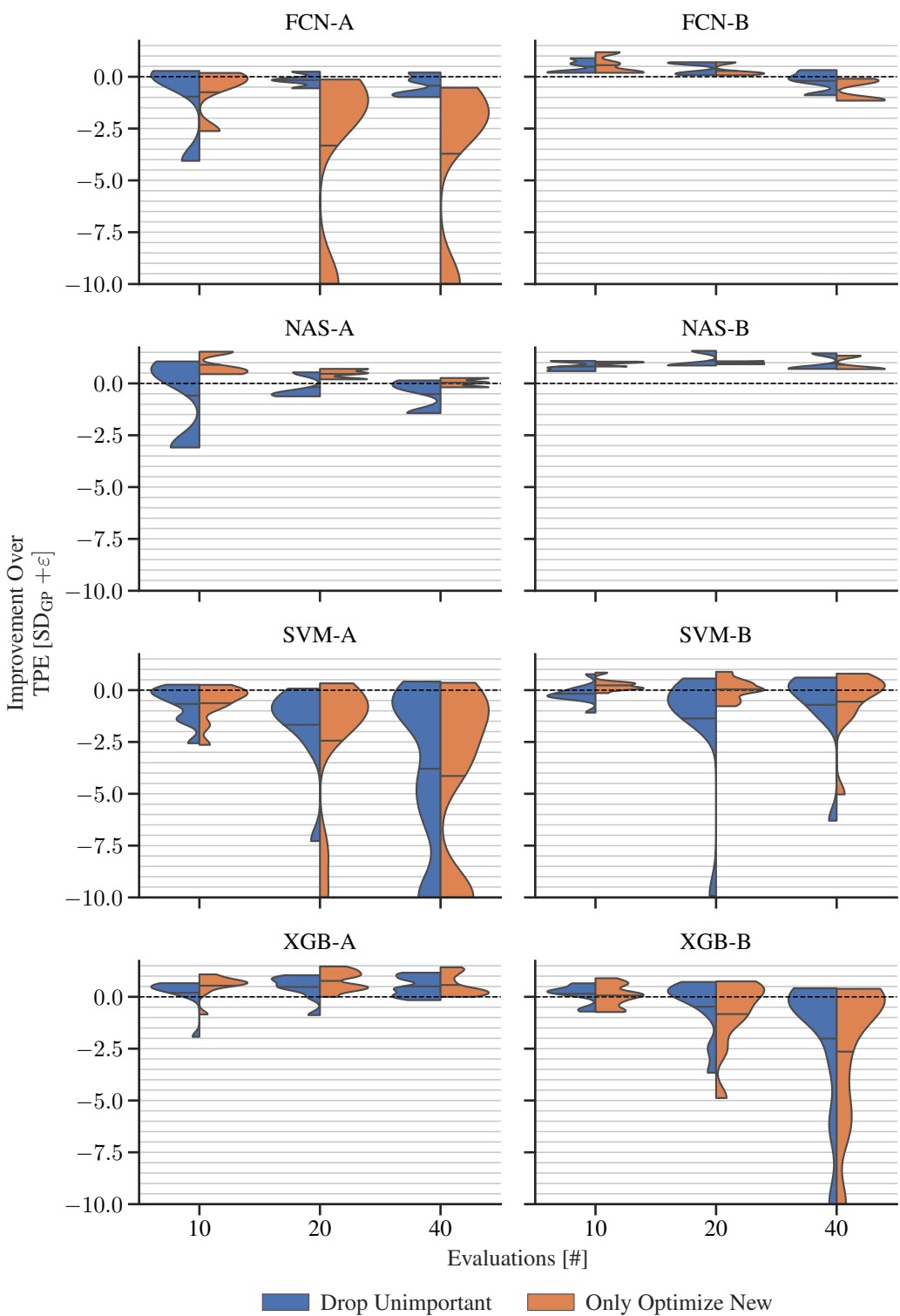

Figure 29: Standardized objective improvements of only-optimize-new and drop-unimportant across tasks for each of 8 benchmarks. The previous HPO has a budget of 40 evaluations.

# E    CONTROL STUDY: GP AND TPE FOR DIFFERENT RANDOM SEED RANGES

As a sanity check, and to gauge the influence of random seeds, we compare GP (and TPE) to itself with different random seed ranges ($GP_1$ and $GP_2$). We observe little differences in $GP_1$ and $GP_2$ (likewise for TPE) compared to the differences between GP and the transfer approaches (Figure 30 and Figure 31).

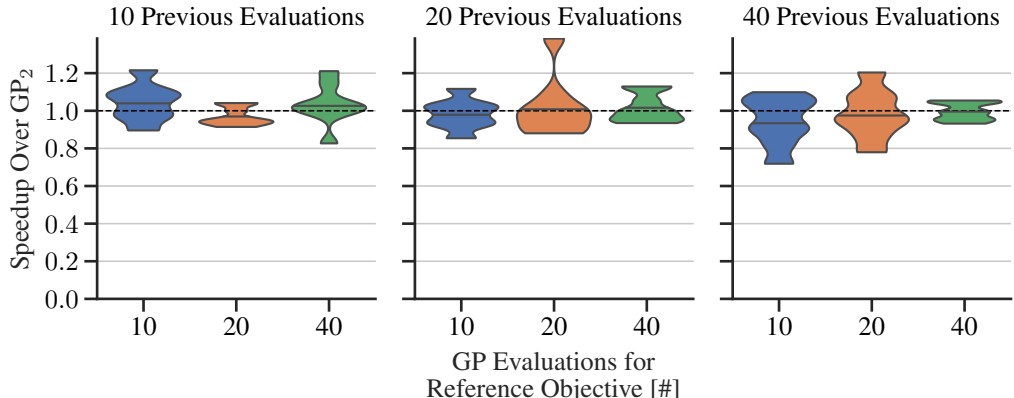

Figure 30: Speedup of $GP_1$ over $GP_2$ across 8 benchmarks. The violins estimate densities of the benchmark geometric means. The horizontal line in each violin shows the geometric mean across these benchmark means. #Evaluations for the old HPO increases from left to right. The x-axis shows the budget for the GP reference.

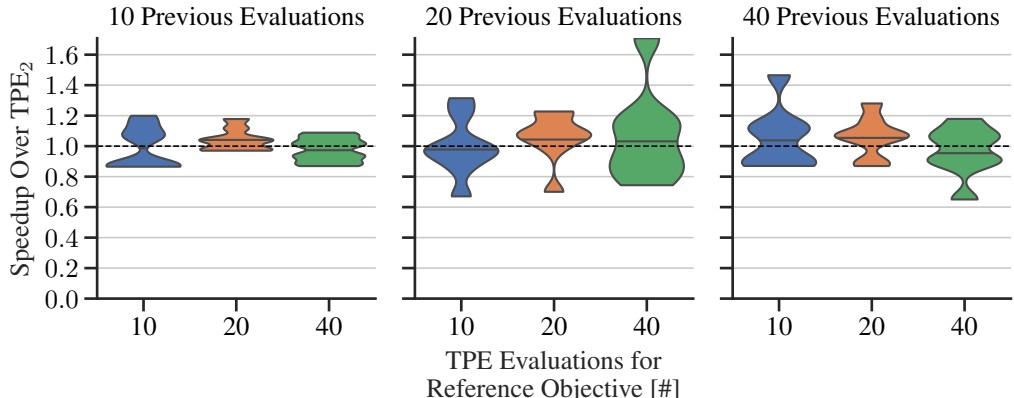

Figure 31: Speedup of $TPE_1$ over $TPE_2$ across 8 benchmarks. The violins estimate densities of the benchmark geometric means. The horizontal line in each violin shows the geometric mean across these benchmark means. #Evaluations for the old HPO increases from left to right. The x-axis shows the budget for the TPE reference.

# F    CONTROL STUDY: RELIABILITY OF GP AND TPE

As a sanity check for the reliability of GP and TPE, we compare them to random search.

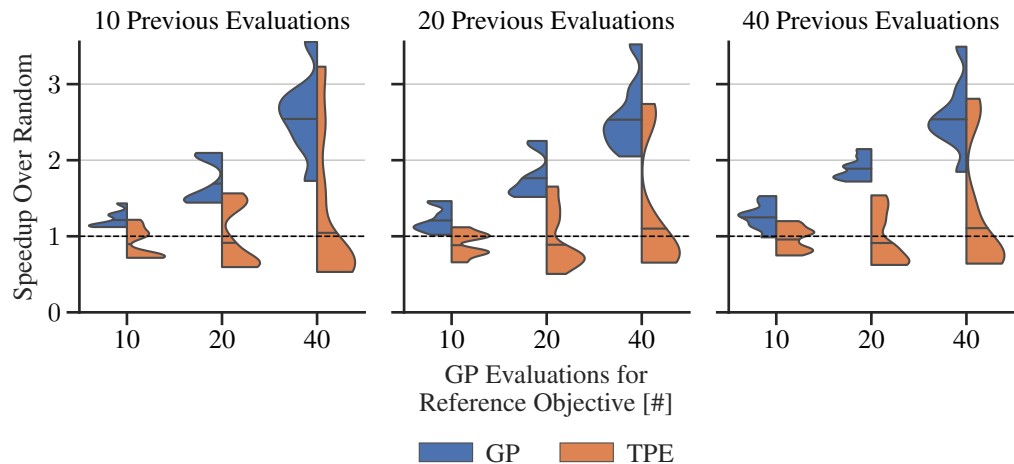

Figure 32: Speedup of GP and TPE over random search across 8 benchmarks. The violins estimate densities of the benchmark means. The horizontal line in each violin shows the geometric mean across these benchmark means. #Evaluations for the old HPO increases from left to right. The x-axis shows the budget for the GP reference.

