# OpenReview forum: "Hyperparameter Transfer Across Developer Adjustments"
_ICLR.cc/2021/Conference — Reject_

### Official Review · AnonReviewer3 · 2020-10-24
**This manuscript introduces a benchmark suite of hyperparameter optimization tasks, that simulate slight developer modifications, with the goal of optimizing a slightly modified algorithm given knowledge of its previous form. The paper further studies the performance of some deliberately naive approaches as a performance benchmark to accompany the suite.**

**Rating:** 5
**Confidence:** 4

**Review:**

Major weaknesses of the paper:
- My understanding is that these are surrogate models that are meant to simulate a real-world task. However there is no description as to how these surrogates were created or trained, nor how their fidelity to the original task was vetted.
- The most simple and naive algorithm seems to provide similar speed-ups to the much more complicated proposed T2PE; especially when considering the much larger improvement of the naive method (see Fig. 11), none of the other proposed methods seem justified to me.
- Furthermore, I don't consider this naive approach (Best First) as being an HPO approach that leverages transfer, since it does exactly what anyone would do when faced with a slightly altered set of hyperparameters.
- This last point suggests that at least one of the following must be true:
  * non-trivial transfer is not as important as intuition would lead us to think;
  * this benchmark suite does not provide a good testbed for assessing an HPO method's ability to transfer; or
  * none of the non-trivial proposed algorithms do a good job transferring and can therefore not argue against the previous point.

Given this important contradiction, I must recommend a rejection. My recommendations would be to:
- focus on creating a good benchmark suite (perhaps focus on a single or two domains as introduce many variants, instead of four domains with only two variants);
- focus on vetting the surrogates in terms of their fidelity to the task they are meant to simulate; and
- focus on demonstrating that accounting for the slight modifications in the subsequent HPO does indeed provide a benefit over the naive thing to do.
For the record, I don't think this is an easy task.

Minor points:
- Justify geometric mean. I'm not saying it's the wrong way to compare these, I just think it requires at least a sentence of justification.
- Same for the violin plots. For such simple plots, simple boxes and whiskers, with perhaps data points to show the spread of measurements across seeds, would do just fine.
- Figure 4, and indeed any mention of the two methods therein, can be entirely removed from the paper; other than to perhaps mention that they were tried and failed---results in the appendix.
- A much more interesting replacement for that figure would be Figure 11.
- Not sure what is the point of comparing random search to TPE in the appendix unless this means Best-first then Random-search/TPE? If the latter is true, please clarify.

---

> ### Author Response · Authors · 2020-11-20
> **Initial Response 2/2**
>
> 3. “The most simple and naive algorithm seems to provide similar speed-ups to the much more complicated proposed T2PE; especially when considering the much larger improvement of the naive method (see Fig. 11), none of the other proposed methods seem justified to me. [...] My recommendations would be to: [...] focus on demonstrating that accounting for the slight modifications in the subsequent HPO does indeed provide a benefit over the naive thing to do. For the record, I don't think this is an easy task.”
>
> --> First, we refer to the reply to all reviewers where we elaborate on the goals of our empirical evaluation, and explain in detail why the empirical evaluation of only-optimize-new and drop-unimportant is quite valuable to the community. Second, we have dropped the range adjustment part of T2PE (in the revised version TGP as we use GPs instead of TPE now, based on the reviews) to make it much simpler. TGP is now like only-optimize-new, but instead of using the best previous values of previously existing hyperparameters at each step, the algorithm samples from a model fitted on the projected results of the previous HPO. Finally, we actually do provide an approach that does indeed yield a benefit over best-first (which is not an easy task!), i.e., the combination of best-first and T2PE (in the revised version TGP). This approach provides an additional 0.1-0.3 average speedup where the number of evaluations for the previous HPO is larger than 10 (Table 2). We made changes to feature this result more prominently and added per-benchmark results for the combination of best-first and TGP to Appendix B (showing that the speedup of TGP+best-first over best-first is up to a factor of 1.5x in some benchmarks; see e.g., Figure 10, FCN-B and NAS-A).
>
>
> 4. “My recommendations would be to: focus on creating a good benchmark suite (perhaps focus on a single or two domains as introduce many variants, instead of four domains with only two variants); [...] “
>
> --> Our benchmark suite (a) is based on code and data from commonly to widely used benchmarks in HPO research, (b) covers a wide range of algorithms, (c) includes developer adjustments of many different types and with many motivations [see our response to AnonReviewer4 point 4(which includes the large appendix comment “Appendix: Motivations for Developer Adjustments”)], (d) includes many tasks for each algorithm and adjustment, and (e) is independent of hardware and comparatively cheap to evaluate. We think these are the attributes of a high quality benchmark suite.
>
>
> 5. “Justify geometric mean. I'm not saying it's the wrong way to compare these, I just think it requires at least a sentence of justification.”
>
> --> We agree and have added a justification to our paper. Intuitively, the geometric mean is an average of speedup values. E.g., two speedups of 0.1x and 10x intuitively average to 1x, and not 5.05x. We want to note that the arithmetic mean is an upper bound for the geometric mean, so using the geometric mean in fact makes our speedups slightly less impressive than had we used the standard arithmetic mean.
>
>
> 6. “Same for the violin plots [justification]. For such simple plots, simple boxes and whiskers, with perhaps data points to show the spread of measurements across seeds, would do just fine.”
>
> --> We agree that we should explain why we use violin plots and have added a justification to the results paragraph in Section 5. The main advantage of using violin plots is being able to take into account potential multi-modality of the data distribution. E.g., in Figure 3 the geometric mean of the best-first approach is not always at a point of high density and the distribution over 8 benchmarks has multiple modes. While stated in the respective captions, here, we also want to note that our plots show either violins over benchmark geometric means, or violins over task geometric means for each benchmark. The seeds are averaged for each task individually.
>
>
> 7. “Figure 4, and indeed any mention of the two methods therein, can be entirely removed from the paper; other than to perhaps mention that they were tried and failed---results in the appendix.”. A much more interesting replacement for that figure would be Figure 11.
>
> -->  In the interest of brevity, we refer to the reply to all reviewers, where we have answered this question in detail.
>
>
> 8. "Not sure what is the point of comparing random search to TPE in the appendix unless this means Best-first then Random-search/TPE? If the latter is true, please clarify."
>
> -->  One reviewer asked how reliable TPE is. A comparison to random search answers this. We now explicitly state this in the description. Please note that we now also include GPs in our evaluation which provide much larger speedups over random search.
>
>
> Thanks again for all your comments! If we cleared up some of your concerns, we would kindly ask you to update your assessment.

---

> ### Author Response · Authors · 2020-11-20
> **Initial Response 1/2**
>
> Thanks for your helpful comments and suggestions.
>
>
> 1. “This manuscript introduces a benchmark suite of hyperparameter optimization tasks, that simulate slight developer modifications, with the goal of optimizing a slightly modified algorithm given knowledge of its previous form. The paper further studies the performance of some deliberately naive approaches as a performance benchmark to accompany the suite.”
>
> --> This summary is not in line with what we present as our main contribution: ‘the introduction of a new research framework: Hyperparameter transfer across adjustments (HT-AA)‘, which is very prominently featured in the abstract and introduction. Our main contribution is not a benchmark suite, but the introduction of a new *problem* that could be as impactful as the problem of hyperparameter transfer across tasks (HT-AT). This distinction is rather important for the reviewing process, as papers that introduce problems, compared to papers that e.g., try to better address a known problem, require different considerations as to potential value and impact. In our reply to all reviewers we discuss this aspect in detail.
>
>
> 2. “My understanding is that these are surrogate models that are meant to simulate a real-world task. However there is no description as to how these surrogates were created or trained, nor how their fidelity to the original task was vetted. [...] My recommendations would be to [...] focus on vetting the surrogates in terms of their fidelity to the task they are meant to simulate; [...]”
>
> --> Half of our benchmarks (4 out of 8) do not use surrogate models but lookup tables. The SVM and XGB benchmarks are surrogate benchmarks and FCN and NAS benchmarks are tabular benchmarks (for an explanation of the difference we refer to Section 4). While in the original submission we provided citations and said that some benchmarks are based on surrogates and some on tabular data, we have now added this information explicitly. Further, all our benchmarks use code and runtime data from existing benchmarks in HPO research. While we have made this clearer in our revision, we would like to note that our original version already mentions this and includes respective citations in Section 4. As for the surrogates, we use an available open-source implementation by the HPOlib authors, which we now explicitly mention in Appendix A. Finally, in the case of the two NAS benchmarks the search space is based purely on categorical hyperparameters (i.e., which operation to apply on a given edge in the architecture graph) and objective values for all potential hyperparameter settings are in the lookup table. Therefore, the two NAS benchmarks are true to HPO for non-simulated code. Qualitatively, the results on the NAS benchmarks are similar to what we see across all benchmarks (Appendix C).
> We believe that the reviewer might have misunderstood the nature of our benchmarks, so let us clarify: regardless of what available open-source benchmark (tabular or surrogate) we based our HT-AA benchmarks on, we never fit a surrogate. We simply use one part of the table (/one part of the surrogate model’s search space) to define one “version 1 of the code” and another part for “version 2 of the code”, and study the transfer from version 1 to version 2.

---

> ### Comment · AnonReviewer3 · 2020-11-23
> **After reading the authors comments and revision**
>
> I see now that the novelty of the paper was not in the suite of surrogates to a benchmark suite of tasks, therefore a few of my comments do not apply. Instead the paper seems to be formalizing a task that most practitioners are very familiar with, namely tuning a black-box after making minor modifications to the algorithm or hardware. I can see some value in such a formalization. Furthermore, having Transfer GP as a good baseline for the benchmark is valuable; although that approach can certainly use a much better explanation.
>
> Overall, I'll increase my score to allow for some discussion.

---

> > ### Author Response · Authors · 2020-11-25
> > **Improved Explanation and Added Pseudocode**
> >
> > Thanks a lot for reading through our response in detail, seeing value in our framework, calling Transfer GP a good and valuable baseline, and finally, for increasing your score.
> >
> > We have improved the explanation of Transfer GP along with other improvements to the approaches section, and added TPE (and Transfer TPE) back in our evaluation on the request of another reviewer. We also added Pseudocode for Transfer GP/TPE in the main paper.
> >
> > Please let us know if you would like further details on Transfer GP, or have any additional suggestions for improving our paper.

---

### Official Review · AnonReviewer4 · 2020-10-26
**Considers problem of warmstarting hyperparameter optimization after small changes to training algorithm and/or HP search space.**

**Rating:** 5
**Confidence:** 4

**Review:**

This paper is addressing a problem which is quite relevant in practice, namely how to warmstart HP optimization after small changes have been done to the ML model. Such changes may modify the HP search space, both by adding/removing HPs, or by changing their value ranges. The paper is clearly written. It introduces 3 potential baselines, as well as a simple transfer strategy. All work is based on TPE, which is frequently used, but not SotA for HPO.

The paper does not elaborate on the motivations of these "developer changes". One could suspect many are attempts to modify/improve the HPO process itself, over the *same* model. And this is not really new, there is lots of prior work to help shaping search spaces, both by quantifying HP relevance or by learning search ranges. Say, a developer modifies the value range of an HP. What other motivation would there be than mistrust in the previous range, but no change of algorithm. Same for adding/removing an HP, which normally just means going from fixed default to HPO or back. In fact, the 8 benchmarks are all of that sort. My feeling is that by viewing the problem in this way (namely, just HP search space optimization), there is suddenly a lot more related work not taken into account here. More difficult problems, such as learning ensembles from a range of models, and then adding/removing model types, are not tackled here. These would call for more difficult transfer strategies.

This paper does not really propose new methodology, except maybe T2PE, which is a pretty basic heuristic. There is a lot of prior work on transfer HPO, some of which could cewrtainly be adopted. Given that the paper is mainly empirical, one would expect a more thorough and wider evaluation. On the positive side, the paper introduces 8 new benchmarks, even though they are pretty simple setups. Their empirical evaluations are a little thin. Only the best-first baseline works well, results for the others are not shown. It should be noted that best-first is standard in HPO practice, this is the first thing one does for transfer. Their T2PE essentially works just as well, and a combination of the two works slightly better. While the paper categorizes types of modification, the empirical evaluation does not differentiate among them anymore. Also, the restriction to TPE is questionable. Why not also use GP-BO? All baselines would work just the same.

My main recommendation for this work would be to be clear about the modification for such limited developer changes. If this is just about the developer trying to twist HPO in itself, this work would have to compare against previous work for optimizing search spaces. Otherwise, please address more complex scenarios, such as ensemble learning, where HPO transfer becomes really difficult.

---

> ### Author Response · Authors · 2020-11-20
> **Appendix: Motivations for Developer Adjustments 2/2**
>
> Heterogeneous adjustments that change X by changing the range for one or multiple hyperparameters (and optionally other changes to X, A, or H):
> * Motivations in the case of numerical hyperparameters could be: optimizing the search space, moving to a GPU with larger memory so that e.g., the batch size or model size can now be larger, changing from a discrete range to a continuous range after being made aware of the possibility of a relaxation of a hyperparameter, etc.
> * Motivations in the case of categorical hyperparameters could be: removing a choice that has a bug, adding a choice after implementing the corresponding code in A (e.g., new type of optimizer or kernel), moving to hardware H that allows for additional choices (e.g., different arithmetic precisions or representations), …
> * The following benchmarks include such adjustments for numerical hyperparameters:
>     * SVM-B (Increase range for cost hyperparameter): E.g., search space optimization.
> * The following benchmarks include such adjustments for categorical hyperparameters
>     * FCN-B (Add per-layer choice of activation function; Change learning rate schedule) E.g., when a developer implements additional activation functions and now wants to optimize over them, and at the same time performs a homogeneous adjustment.
>     * NAS-A (Add 3x3 average pooling as choice of operation to each edge): E.g., after learning about the idea of average pooling, the developer adds this to the NAS search space.
>
> Heterogeneous adjustments that change X by adding or removing one or multiple hyperparameters (and optionally other changes to X, A, or H):
> * Motivations could be: unfixing/exposing an existing hyperparameter, fixing an existing hyperparameter like the batch size after moving to a GPU with less memory, fixing an existing categorical hyperparameter because a bug has been found in one of two choices, a part of the algorithm A was changed and the new version of that part includes one or multiple new hyperparameters (e.g., when changing SGD to ADAM, when updating to a GPU that has support for low precision arithmetics, or when changing the NAS cell template), a part of algorithm A was changed and the new version of that part does not include certain hyperparameters anymore.
> * The following benchmarks include such adjustments:
>     * FCN-A (Increase #units-per-layer 16×; Double #epochs; Fix batch size hyperparameter) E.g., a developer received a more powerful GPU, and could hence increase the model size and training time, but had to fix the batch size to fit the larger model into GPU memory.
>     * FCN-B (Add per-layer choice of activation function; Change learning rate schedule): E.g., when a developer implements additional activation functions and now wants to optimize over them, and at the same time performs a homogeneous adjustment.
>     * XGB-A (Expose four booster hyperparameters): The developer learns that certain hyperparameters are important to tune, and hence does not use the default values of the library anymore.
>     * NAS-B (Add node to cell template (adds 3 hyperparameters)). E.g., when moving to a larger GPU that can fit a larger neural network into memory.
>     * SVM-A (Change kernel; Remove hyperparameter for old kernel; Add hyperparameter for new kernel). E.g., when a visualization of the data clearly shows that a radial kernel makes no sense, but a polynomial kernel does, however, now the degree of the polynomial needs to be tuned (which did not exist as a hyperparameter before), and the hyperparameter for the radial kernel is dropped.
>
> We hope that this extensive list demonstrates the broad applicability of HT-AA, and that it is by no means limited to search space optimization. If this broad applicability of our problem formulation changes a reviewer’s mind, we would kindly ask them to update their score accordingly.

---

> ### Author Response · Authors · 2020-11-20
> **Appendix: Motivations for Developer Adjustments 1/2**
>
> First, let us introduce some notation: The objective f(A(x), T, H) that the hyperparameter optimization algorithm tries to optimize depends on three entities: the learning algorithm A(x) instantiated with hyperparameters x \in X, the task T (in supervised learning this would be a train- and validation/test dataset and the evaluation metric), and the hardware H the algorithm is run on.
> In our paper we differentiate between heterogeneous and homogeneous developer adjustments: Heterogeneous adjustments change X and potentially also A or H; while homogeneous adjustments do not change X, but change at least one of A or H. In the following we further differentiate these adjustment types, to then discuss motivations and list which benchmarks in our benchmark suite are examples of a given type.
>
> Homogeneous adjustments that change A (and optionally H):
> * These could change any arbitrary part of algorithm A, as long as this change does not affect the search space X.
> * Motivations could be: fixing a bug, adding support for different hardware H (e.g., TPU, different robotic embodiment, …), enabling use of special hardware subroutines, rewriting the complete implementation, changing any constants in the code, improving or changing any part of the learning model, optimization routine, or dataloading, etc. .
> * The following benchmarks include such adjustments:
>     * FCN-A (Increase #units-per-layer 16×; Double #epochs;Fix batch size hyperparameter). A scenario for this benchmark could e.g., be that a developer received a more powerful GPU, and could hence increase the model size and training time, but had to fix the batch size to fit the larger model into GPU memory.
>     * FCN-B (Add per-layer choice of activation function; Change learning rate schedule). A scenario for this benchmark could e.g., be that a developer has finally implemented a learning rate schedule that is known to perform good for the task T, and at the same time performs a heterogeneous adjustment.
>     * XGB-A (Change four unexposed booster hyperparameter values). This could occur e.g., when a developer changes the default values of a used library to settings used in the literature.
>     * SVM-A (Change kernel; Remove hyperparameter for old kernel; Add hyperparameter for new kernel). E.g., when a visualization of the data clearly shows that a radial kernel makes no sense, but a polynomial kernel does.
>
> Homogeneous adjustments that change H (and optionally A):
> * These could change any part of the hardware the algorithm is run on
> * Motivations could be: Change the robotic embodiment to use stronger actuators, change the material of the tires of the robot, use a more powerful CPU to preprocess the data, run on a bigger number of GPUs and change the data parallelisation mode, increase RAM to avoid loading bottlenecks, move to TPU, change to a GPU with hardware subroutines for sparse neural networks, change to a GPU with hardware subroutines for low precision arithmetics, etc.
> * We do not include any hardware adjustments in our benchmarks, since we did not have access to different hardware environments, but we are committed to doing so in the future. A concrete path to this just appeared by means of a new hardware-aware NAS benchmark HW-NAS-Bench (a parallel ICLR submission: https://openreview.net/forum?id=EohGx2HgNsA), which would allow changing between six different hardware platforms (note that the paper also shows that different architectures have optimal tradeoffs on different hardware, i.e., the architectural hyperparameters need to change after a change of hardware).

---

> > ### Comment · AnonReviewer4 · 2020-11-23
> > **No change**
> >
> > Your response contains a lot of "could", while all you are doing in these benchmarks is to modify the search space of hpo. I hope you understand what my concern is.
> > If you want to do something new, then please do it and do not just suggest it is being done. Create more convincing benchmarks, where real developer changes are happening. Do not just suggest that changes to the search space can be interpreted this way.

---

> > > ### Author Response · Authors · 2020-11-23
> > > **We Reply Above**
> > >
> > > We disagree and have replied to your response above (https://openreview.net/forum?id=WPO0vDYLXem&noteId=ViaG_ysLD-y).

---

> ### Author Response · Authors · 2020-11-20
> **Initial Response 2/2**
>
> Unbounded HPO: The literature on unbounded HPO (some citations below) does not take a transfer into account, and therefore assumes an already evaluated hyperparameter setting to not change in objective value (other than perhaps noise). This is problematic, as homogeneous and heterogeneous adjustments can change any part of the algorithm A or hardware H, and hence change the value of f(A(x), T, H) without changing x. Also, the concept of unbounded HPO does not apply to categorical hyperparameters (there are no bounds), of which we have many in our experiments and in representation learning in general.
>
> Shahriari, B., Bouchard-Côté, A., & Freitas, N. (2016, May). Unbounded Bayesian optimization via regularization. In Artificial intelligence and statistics (pp. 1168-1176).
>
> Ha, H., Rana, S., Gupta, S., Nguyen, T., & Venkatesh, S. (2019). Bayesian Optimization with Unknown Search Space. In Advances in Neural Information Processing Systems (pp. 11795-11804).
>
> Nguyen, V., Gupta, S., Rana, S., Li, C., & Venkatesh, S. (2019). Filtering Bayesian optimization approach in weakly specified search space. Knowledge and Information Systems, 60(1), 385-413.
>
> If you are satisfied with these elaborations we would be happy to include parts of it in the paper and/or appendix. Would that be fine?
>
>
> 3. “This paper does not really propose new methodology, except maybe T2PE, which is a pretty basic heuristic. There is a lot of prior work on transfer HPO, some of which could certainly be adopted. Given that the paper is mainly empirical, one would expect a more thorough and wider evaluation.
>
> --> We want to note that while there is a lot of prior work on HPO transfer across tasks, we are the first to draw attention to the problem of hyperparameter transfer across adjustments (the task does not change but the algorithm, hardware, and/or search space). A paper with the main goal of drawing attention to a new problem requires a different empirical treatment compared to a paper that better addresses a known problem. In our reply to all reviewers above, we elaborate on the goals of our empirical evaluation. As for adapting existing algorithms for hyperparameter transfer across tasks to this new setting, (or even for a transfer across tasks and across adjustments), we see this as an exciting future research direction -- one among many that our paper gives rise to.
>
>
> 4. “Only the best-first baseline works well, results for the others are not shown.”
>
> -->  This must have been a misunderstanding, as we do show results for all methods we describe: We compare T2PE (in the revised version TGP as we use GPs instead of TPE now, based on the reviews) with best-first in Figure 3, and T2PE with best-first+T2PE in Table 2. Violin plots for best-first+T2PE could be found in Appendix D (now in the main paper). For drop-unimportant and only-optimize-new we can not show speedups, as these approaches fail to reach the target objective in a very large percentage of cases. Instead we showed these failure rates in Figure 4 (now Figure 5).
>
>
> 5. “While the paper categorizes types of modification, the empirical evaluation does not differentiate among them anymore.”
>
> -->  We added a differentiation to Table 1. For the analysis itself we do not differentiate among the categories though, as many benchmarks have multiple categories of adjustments. Having said that, we want to point out that in Appendix B, we show performance on a per-benchmark-basis.
>
>
> Thanks again for all your comments! If we cleared up some of your concerns, we would kindly ask you to update your assessment.

---

> ### Author Response · Authors · 2020-11-20
> **Initial Response 1/2**
>
> Thanks for your comments, suggestions, and for recognizing the practical relevance of the proposed problem setting. In the following we provide detailed replies to your questions and comments.
>
>
> 1. “All work is based on TPE, which is frequently used, but not SotA for HPO. [...] Why not also use GP-BO?”
>
> --> We have replaced TPE with GP for all transfer and non-transfer approaches. This update results in a much faster non-transfer baseline compared to TPE, and at the same time, the GP based transfer approaches provide a larger speedup over GP than the TPE based transfer approaches did over TPE, leading to an overall much larger speedup. Thank you for the proposal, we think this made the paper a lot stronger!
>
>
> 2. “The paper does not elaborate on the motivations of these "developer changes". One could suspect many are attempts to modify/improve the HPO process itself, over the same model. And this is not really new, there is lots of prior work to help shaping search spaces, both by quantifying HP relevance or by learning search ranges. Say, a developer modifies the value range of an HP. What other motivation would there be than mistrust in the previous range, but no change of algorithm. Same for adding/removing an HP, which normally just means going from fixed default to HPO or back. In fact, the 8 benchmarks are all of that sort. My feeling is that by viewing the problem in this way (namely, just HP search space optimization), there is suddenly a lot more related work not taken into account here.”
>
> --> As we think this is an important discussion and you have signaled that this is your main concern, we will answer this question on the possible motivations of developer changes with great detail. As there are so many potential motivations for developer adjustments, we created a separate comment thread that serves as an appendix and answers your question by listing motivations for each adjustment type we differentiate, which of our benchmarks include a given type of adjustment, and example motivations for the adjustments of each of our benchmark. In addition to this appendix, in this main response, we relate our framework (HT-AA) to search space learning and unbounded hyperparameter optimization.
>
>
> First, let us introduce some notation: The objective f(A(x), T, H) that the hyperparameter optimization algorithm tries to optimize depends on three entities: the learning algorithm A(x) instantiated with hyperparameters x \$\in$ X, the task T (in supervised learning this would be a train- and validation/test dataset, and the evaluation metric), and the hardware H the algorithm is run on. As a reminder, in our paper we differentiate between heterogeneous and homogeneous developer adjustments: Heterogeneous adjustments change X and potentially also A or H; while homogeneous adjustments do not change X, but change at least one of A or H.
>
> Learning search spaces: approaches like the one of Perrone et. al. (2019) learn to prune X by using a meta training set of tasks {T}, i.e., they are approaches for hyperparameter transfer across tasks (HT-AT). As such they are evaluated using a large number of meta training tasks. These could in principle be adapted for HT-AA, by pruning only the part of the search space that was already present before the developer adjustment, similarly to how T2PE (in the revised version TGP as we use GPs instead of TPE now, based on the reviews) builds a model only for the already existing hyperparameters. However, in the basic HT-AA problem we only have data for one task T for one development step, i.e., only one previous run in general. This is problematic, as these approaches might prune parts of the search space that are only good after the developer adjustments and would have no way to correct this (Perrone et. al. (2019); Wistuba et. al. (2015) for categoricals). Applying e.g., the approach of Perrone et. al. (2019) with the adaptation to HT-AA described above, to a search space of continuous hyperparameters across a homogeneous adjustment, is equivalent to only-optimize-new (which performs horribly in our experiments). The above discussion also highlights some of the issues in adapting approaches designed for a transfer across tasks (HT-AT) to a transfer across developer adjustments (basic HT-AA).
>
> Perrone, V., Shen, H., Seeger, M. W., Archambeau, C., & Jenatton, R. (2019). Learning search spaces for Bayesian optimization: Another view of hyperparameter transfer learning. In Advances in Neural Information Processing Systems (pp. 12771-12781).
>
> Wistuba, M., Schilling, N., & Schmidt-Thieme, L. (2015, September). Hyperparameter search space pruning–a new component for sequential model-based hyperparameter optimization. In Joint European Conference on Machine Learning and Knowledge Discovery in Databases (pp. 104-119). Springer, Cham.

---

> > ### Comment · AnonReviewer4 · 2020-11-23
> > **Read author response**
> >
> > I read the verbose author response, thanks for putting this together.
> > I have to say I am still not convinced. The authors seem to claim this is a novel setup, and I just do not see a proper justification. Their use cases are instances of modifying the search space, and there is enough prior work on that one. They cannot avoid having to compare against such work by just claiming what they do is novel and different.
> >
> > They have two options. Either recognize this is more or less an instance of modifying the search space plus some transfer, and then provide a proper comparison. Or come up with much more convincing benchmarks, that would really convince me of them not just being an instance of changing the search space.
> >
> > No change in my vote.

---

> > > ### Author Response · Authors · 2020-11-23
> > > **Our benchmarks do not only modify the search space**
> > >
> > > Thank you for reading our detailed response and your further comments.
> > >
> > > No, our benchmarks do not only modify the search space X (see Table 1; adjustments labelled as homogeneous do not change the search space). E.g., in XGB-B (which changes four constants) the search space does in fact not change at all across our simulated developer adjustment. Most other benchmarks include developer adjustments that do not change the search space, e.g., in FCN-A (which among other changes increases the number of epochs), or FCN-B (which among other changes includes the change to a different learning rate schedule).
> > >
> > > Is your issue that we simulate developer changes by defining a larger search space and then define one part of the search space as “version 1” and a different part of the search space as “version 2” (in addition to e.g., changing the #epochs)? If so, any changes to an algorithm can be seen this way.
> > >
> > > Also, in our initial reply we already relate to search space learning and unbounded Bayesian optimization: We discuss why for our benchmarks search space learning (Perrone et al, 2019) modified to work with heterogeneous adjustments, is equivalent to only-optimize-new (except that that technique is only defined on continuous dimensions and simply ignores categoricals); we therefore already do provide that comparison for benchmarks without categorical hyperparameters in the initial HPO. We also discuss why unbounded Bayesian optimization does not apply to our benchmarks (there is no transfer learning and categoricals have no bounds). Unbounded BO is an alternative to numerical range adjustments, not an alternative to HT-AA algorithms.
> > >
> > > We would very much like to provide a comparison against any method the reviewer suggests, if feasible even during the limited time remaining in the author response period, and more comprehensively for the final version, and we are standing by to hear which one we should compare to.

---

### Official Review · AnonReviewer1 · 2020-10-28
**Valuable framework, precisions required**

**Rating:** 6
**Confidence:** 4

**Review:**

The authors propose a new framework for hyperparameter optimization and transfer across incremental modifications to a given algorithm and its search space, a process called developer adjustments in the paper. The authors then propose a few strategies to transfer knowledge from previous HPO runs and evaluate them on a series of simulated benchmarks. Results show the value added by transferring information from previous runs, as well as the surprising efficiency of simply reusing the best found hyperparameters from the previous run.

Strong points:

- The framework is simple and clearly introduced.
- Extensive experiments help bring to light the advantage of transferring across adjustments.
- The paper is very well written.

Weak points:

- Not enough details on benchmarks, more on this below
- The use of simulated benchmarks with surrogate models introduces noise in the evaluation
- Comparisons with more baselines would be beneficial. RF and/or GP-based HPO methods are extremely popular and would have been easy to integrate with the best-first baseline.

Recommendation:

The contributions are simple and incremental, and clearly rooted in machine learning engineering, however I still think they could be beneficial as a whole to the community given the extensive experiments realized. I have some issues with experiments, lack of details and baselines, but those issues are mostly fixable. I'll give the paper a weak accept for now.

Extra comments:

You do not specify which benchmarks are based on lookup tables and which ones are based on surrogate models. From looking at the search spaces, I would assume that the SVM and XGB benchmarks are modeled via surrogate benchmarks and the FCN and NAS benchmarks are lookup tables, but this should be explicited in the paper (or appendix). Parameters used for the benchmark surrogate model should also be given (if defaults of Eggensperger are used, simply mention this). It is also not clear what underlying datasets are used, this bears some importance and should be mentioned, even if only in the Appendix.

On surrogate model benchmarks: It can be seen in (Eggensperger et al. 2015., Figure 2) that ordering of methods can shift due to noise in the surrogate model (a random forest?). This is likely going to have a bigger impact when trying to measure the speedup, which is measured when a method reaches a certain threshold of performance. This threshold is likely to be met during the convergence phase of algorithms, and this phase appears noisier  (i.e. looking at how the phases of transition differ between the true benchmark and the RF surrogate benchmark differ in Eggensperger et al. 2015). Have you given this any thought? Have you compared experiments with a few runs on a real benchmark?

The method you end up recommending only has its detailed performance shown in the appendix. This feels counterintuitive to me. This result should be featured in the paper itself. This is perhaps due to the used of those split violin plots, which force you to display only two methods per plot. Maybe you should display a group of X single-sided violin plots where X is the number of methods you are trying to compare.

I think it is misleading to portray everything in terms of speedup or improvement over the "TPE solution with X iterations". A more strictly meaningful metric here is accuracy (assuming there is only one dataset per benchmark). Assuming the performance to beat by original TPE was an 11% error rate, there is a big difference between a method which was able to achieve a 10% error rate and a method which was able to achieve a 5% error rate, yet both will be assessed by how quickly they achieved x < 10% error rate. I can't seem to find such figures in the appendices.

Typos:

- Section 3.1 page 3, argmax g(x) / b(x) << you mean g(x) / l(x)?
- appendix G, you wrote TPE2 instead of T2PE

---

> ### Author Response · Authors · 2020-11-20
> **Initial Response 2/2**
>
> 5. “The contributions are simple and incremental, and clearly rooted in machine learning engineering, however I still think they could be beneficial as a whole to the community given the extensive experiments realized.”
> -->  Thank you for the characterization as beneficial as a whole to the community. However, we consider our contributions not as incremental, as we introduce a practically relevant and fundamentally novel research problem.
>
>
> 6. “The method you end up recommending only has its detailed performance shown in the appendix. This feels counterintuitive to me. This result should be featured in the paper itself.”
> --> We agree. We changed the paper to feature the recommended method more prominently.
>
>
> 7. “I think it is misleading to portray everything in terms of speedup or improvement over the "TPE solution with X iterations". A more strictly meaningful metric here is accuracy (assuming there is only one dataset per benchmark). [...] I can't seem to find such figures in the appendices.”
> --> We consider multiple datasets per benchmark (see Table 3 in Appendix A) and our benchmarks even use different metrics (now also in Table 3), so we chose speedup as a metric to bridge all these. Additionally, besides the speedup evaluation, we also provide an evaluation of standardized objective improvement over the "TPE/GP solution with X iterations" (with a small delta added, as some standard deviations were 0) in Appendix D (referenced at the end of the experiments section). While this type of evaluation is more common in research on the related hyperparameter transfer across tasks problem, in the main paper we decided to focus on the speedup instead, as the viewpoint of reducing computational demands is ethically stronger (reducing the carbon footprint, etc), and as speedup is easier to interpret than some normalized average objective improvement with a small delta added.
>
>
> 8. “Appendix G, you wrote TPE2 instead of T2PE”
> --> Actually, this is not a mistake. In Appendix G we compare TPE with two different seed ranges (denoted TPE and TPE2) to measure the influence of seeds in our evaluation. But we realized that this may have been confusing and renamed the two seed ranges to TPE_1 and TPE_2.
>
>
> Thanks again for all your comments! If we cleared up some of your concerns, we would kindly ask you to update your assessment.

---

> ### Author Response · Authors · 2020-11-20
> **Initial Response 1/2**
>
> Thanks for the insightful comments and questions, and the suggested improvements! Thanks also for the positive feedback on our writing, the clear introduction of the new HT-AA framework, our extensive experiments, and the value HT-AA provides. We would like to comment on your suggestions, comments and questions in the following.
>
>
> 1. “Comparisons with more baselines would be beneficial. RF and/or GP-based HPO methods are extremely popular and would have been easy to integrate with the best-first baseline.”
> -->  We have replaced TPE with GP for all transfer and non-transfer approaches. This update results in a much faster non-transfer baseline compared to TPE, and at the same time, the GP-based transfer approaches provide a larger speedup over GP than the TPE-based transfer approaches did over TPE, leading to an overall much larger speedup. Thank you for the proposal, we think this makes the paper much stronger.
>
>
> 2. “You do not specify which benchmarks are based on lookup tables and which ones are based on surrogate models. From looking at the search spaces, I would assume that the SVM and XGB benchmarks are modeled via surrogate benchmarks and the FCN and NAS benchmarks are lookup tables, but this should be explicited in the paper (or appendix). Parameters used for the benchmark surrogate model should also be given (if defaults of Eggensperger are used, simply mention this).”
> -->  Yes, SVM and XGB benchmarks are surrogate benchmarks and FCN and NAS are lookup tables. We have added this information to Section 4 of the paper. All our benchmarks use code and runtime data from existing benchmarks in HPO research. While we have made this clearer in our revision, we would like to note that our original version already mentions this and includes respective citations in Section 4. As for the surrogates, we use an open-source implementation by the HPOlib authors, which we now explicitly mention in Appendix A.
>
>
> 3. “It is also not clear what underlying datasets are used, this bears some importance and should be mentioned, even if only in the Appendix.”
> --> We agree and have added this information to Appendix A.
>
>
> 4. “The use of simulated benchmarks with surrogate models introduces noise in the evaluation. [...] Have you compared experiments with a few runs on a real benchmark?”
> -->  Half of our benchmarks (4 out of 8) do not use surrogate models but lookup tables. Additionally, our surrogate benchmarks are modifications of commonly used benchmarks in hyperparameter transfer across tasks. Without the use of simulated benchmarks, research on HPO and especially transfer scenarios for HPO can be prohibitively expensive for underrepresented groups with poor compute resources, and using simulated benchmarks is a standard practice to avoid this. The use of simulated benchmarks is particularly important in our case, as we need to establish a solid foundation for further research, which includes providing cheap and hardware-independent benchmarks. Having said that, in the case of the two NAS benchmarks the search space is based purely on categorical hyperparameters (i.e., which operation to apply on a given edge in the architecture graph) and objective values for all potential hyperparameter settings are in the lookup table. Therefore, the two NAS benchmarks are equivalent (but cheaper) to HPO runs on real code. Qualitatively, the results on the NAS benchmarks are similar to what we see across all benchmarks (Appendix B).

---

> > ### Comment · AnonReviewer1 · 2020-11-24
> > **response to rebuttal**
> >
> > I would like to thank the authors for a very detailed rebuttal. I have read everything, and I have some further comments (for now or future revisions of the paper).
> >
> > - It is nice that you replaced TPEs with the more widely adopted (or state of the art?) GPs, but merely replacing one baseline by the another does not qualify as "increasing the number of baselines".
> >
> > - I am not convinced by your argument on surrogate benchmark usage. I know that they only represent half of the benchmarks, yet the point is that they might present a risk of returning a false result. It doesn't matter if a benchmark suite is more widely available due to lower computing costs if it is at risk of returning incorrect results. A couple of non-simulated experiments could help persuade a reader that this is not the case, and the computing budget of training SVMs of low and mid-sized datasets is far from being prohibitive.
> >
> > - On speedup & normalized average objective improvement: I would say the normalized average objective improvement is also a little misleading, and it abstracts away a lot of information. Perhaps more details should be provided as to how it was computed (or a citation provided for the reader unfamiliar with glass delta), for instance you show the distribution (over repetitions?) of the average improvement on the 8 benchmarks. What is the distribution of the violin plot computed, and what about the mean of the new algorithm? Also once again here, as in many other places in the paper, you fall victim to your choice of two-sided violin plots which only allow for the comparison of two methods in one graph.

---

> > > ### Author Response · Authors · 2020-11-24
> > > **Further Revisions (Major and Minor)**
> > >
> > > Thank you for reading our rebuttal very carefully and for responding quickly with another round of helpful comments.
> > >
> > >
> > > 9. “It is nice that you replaced TPEs with the more widely adopted (or state of the art?) GPs, but merely replacing one baseline by the another does not qualify as "increasing the number of baselines".”
> > >
> > > --> We agree that the results we had for TPE still remain useful and have now added the evaluation with TPE and T2PE into Appendix B, Appendix C, and Appendix D, and have included them as a second row in Figure 3 and Table 1. We hope that this addresses your concern.
> > >
> > >
> > > 10. “I am not convinced by your argument on surrogate benchmark usage. I know that they only represent half of the benchmarks, yet the point is that they might present a risk of returning a false result. It doesn't matter if a benchmark suite is more widely available due to lower computing costs if it is at risk of returning incorrect results. A couple of non-simulated experiments could help persuade a reader that this is not the case, and the computing budget of training SVMs of low and mid-sized datasets is far from being prohibitive.”
> > >
> > > --> We hear your concerns. While we would like to mention that we don’t see surrogates as returning false results but rather as defining a slightly different blackbox function that still shares many properties with the non-surrogate version (much more so than other blackbox functions popular in HPO, like Branin), we do agree that adding some experiments with real benchmarks will increase trust in our empirical results. We are happy to replace the SVM and XGB surrogate benchmarks with similar SVM and XGB benchmarks that do not rely on surrogates for the final version of the paper. In order to keep compute requirements low (also for future use of the benchmarks), we would reduce the number of tasks from 10 to 3 and the number of random seeds from 25 to 15. Would this fix your concerns? Or would you rather have us *add* these non-surrogate benchmarks, rather than replacing the surrogate versions? Either would be fine for us.
> > >
> > >
> > > 11. “On speedup & normalized average objective improvement: I would say the normalized average objective improvement is also a little misleading, and it abstracts away a lot of information.”
> > >
> > > --> We have 54 tasks (with 3 tasks per benchmark this would still be 24 tasks) some of which have different underlying metrics, for each we have multiple repeats, and we look at these over 9 different budget combinations. We are aware of four different approaches in the literature to this kind of data situation: ranking based, standardized improvement, normalized regret, and speedup. We would argue that aggregations (at different levels) of speedup and standardized objective improvement are the most informative of these. In order to guard against losing information by the aggregation across benchmarks, we also provide results for individual benchmarks for speedup in Appendix B and have added this for the improvement in Appendix D. Do you know a better approach to analyzing this data? If so, we would be happy to implement it for the final version of the paper.
> > >
> > >
> > > 12. “Perhaps more details should be provided as to how it was computed (or a citation provided for the reader unfamiliar with glass delta), for instance you show the distribution (over repetitions?) of the average improvement on the 8 benchmarks. What is the distribution of the violin plot computed, and what about the mean of the new algorithm?”
> > >
> > > --> We have added an explanation of how the metric is computed for each task, and details for how we aggregate (same as for the speedup plots) to Appendix D. We now provide results for two aggregation levels: the distribution over task standardized mean improvements, where we compute the standardized mean improvement over repetitions and show results on a per-benchmark level; and the distribution over benchmark means, i.e., the means across task standardized mean improvements for all tasks in a given benchmark.
> > >
> > >
> > > 13. “Also once again here, as in many other places in the paper, you fall victim to your choice of two-sided violin plots which only allow for the comparison of two methods in one graph.”
> > >
> > > --> Thank you for bringing this up again. We have modified our violin plots to show multiple methods. Thank you for the helpful suggestion, we believe this makes the visual presentation of the results a lot stronger.

---

### Official Review · AnonReviewer2 · 2020-10-28
**A framework for hyperparemeter transfer when ML algorithm changeshm**

**Rating:** 5
**Confidence:** 4

**Review:**

The paper is motivated by the situation where a machine learning algorithm has development adjustment and we would like to reuse the tuning results of previous hyperparameter optimization. The paper calls it HT-AA problem, which certainly is an interesting problem given software can get updated often. The paper proposes four simple baseline algorithms for the HT-AA problem.

For the empirical study, a set of eight benchmarks for basic HT-AA problem are presented. The experiment results show the transfer TPE (T2PE) and best-first strategy produce good speed up. To reach given objective values, T2PE can be 1.0–1.7x faster than TPE, and best-first 1.2–2.6x faster comparing with old HPO.

The pros of the paper include:
1. Although the topic of the paper is not one of the most popular, some readers might find it interesting and can be benefited from it.
2. The proposed methods are reasonable and acceptable.
3. Numerical results show some of the proposed methods can help to speed up reoptimizing hyperparameter.

The cons include:
1. Need explain when the only-optimize-new and drop-unimportant methods can be useful. If not useful as the experiments demonstrate, why propose them?
2. Look like TPE is the method that show speedup for the benchmarks. How reliable the method is? Is the saving justifying use an extra tuning tool?

---

> ### Author Response · Authors · 2020-11-20
> **Initial Response**
>
> Thanks for your helpful comments, questions, and the positive feedback for our approaches and numerical results, calling our proposed problem setting interesting, and recognizing the potential benefit our paper can bring to ICLR readers. In the following we provide detailed replies to your questions.
>
>
> 3. “Need explain when the only-optimize-new and drop-unimportant methods can be useful. If not useful as the experiments demonstrate, why propose them?”
>
> --> In the interest of brevity, we refer to the general reply to all reviewers above, where we have answered this question in detail.
>
>
> 4. “Look like TPE is the method that show speedup for the benchmarks. How reliable the method is? Is the saving justifying use an extra tuning tool?”
>
> --> Based on a suggestion by other reviewers, we added Gaussian processes approaches which perform better than TPE. To measure how reliable the basic HPO without transfer is, we compare it to random search in Appendix F. As the cost for HPO scales with the cost of the algorithm it is tuning, an average speedup of 1.2--3.6x (depending on the budgets involved) and a maximum benchmark-average speedup of over 10x can be a significant cost and CO2 saver. As our transfer algorithms fold into the basic hyperparameter tuning tool itself, and do not add any additional required user actions (other than optionally choosing where to save results), using the across-adjustments transfer algorithms does not add any overhead for the user compared to using a tool for basic HPO. As for the maintainer of the tool: the complexity of the approaches we evaluated is deliberately chosen as low, so implementing these approaches is certainly feasible. The simple code for all our tools is open-sourced.
>
>
> Thanks again for all your comments! If we cleared up some of your concerns, we would kindly ask you to update your assessment.

---

> > ### Comment · AnonReviewer2 · 2020-11-23
> > **Author's responses**
> >
> > I appreciate the authors' detailed response of my questions. My concerns are mitigated a bit by the new revision of the paper that shows better results/performance than the original paper. I  think this is an interesting research topic, however, I'd like to see the paper highlights more its differences with prior studies and contributions to this research area. I keep my score unchanged.

---

> > > ### Author Response · Authors · 2020-11-23
> > > **Happy to Include More Discussion**
> > >
> > >
> > > Thank you for your kind words and for your additional suggestion. We are currently assessing where this additional discussion would be best placed. We already have a large related work section (almost 2 pages), but we would be very happy to include additional discussions and elaborations you see as missing in the remaining author response time. We also realize that our related work section comes relatively late in the paper, and we are wondering whether your concern perhaps would best be addressed by featuring the comparison to existing hyperparameter transfer work more prominently, i.e., already in Section 2 instead of in the related work section? We’ll gladly implement the changes you suggest.

---

> > > ### Author Response · Authors · 2020-11-25
> > > **Performed Revision**
> > >
> > > Since we did not hear back anymore in the limited time, we went ahead and implemented the change we proposed below (https://openreview.net/forum?id=WPO0vDYLXem&noteId=6H4FaCc8ozb). If you have any further suggestions, we would be happy to add additional discussion for the final version of the paper.
> > >
> > > Regarding the change: We now provide a detailed discussion on the differences and contributions of HT-AA compared to existing work on hyperparameter transfer already in Section 2. Thank you for your suggestion, we think this earlier relation makes our paper stronger.

---

### Author Response · Authors · 2020-11-20
**Paper Revisions After Reviews**

We performed the following major revisions in response to the reviews:

* Based on the reviewers’ comments, we now base our evaluation on Gaussian processes (GPs). The HPO baseline now uses GPs and all transfer approaches do as well. We refer to the GP-based version of Transfer TPE (T2PE) as Transfer GP (TGP). Since GPs work much better on these low-dimensional problems, this modification results in a much faster non-transfer baseline compared to TPE; at the same time, the GP-based transfer approaches provide a larger speedup over GP than the TPE based transfer approaches did over TPE, leading to an overall much larger speedup. We believe this makes the paper much stronger and thank the reviewers for the suggestion.
* We have dropped the range adjustment part of what was formerly T2PE (now TGP) to make it as simple and easy-to-implement as our other baselines. TGP is now similar to only-optimize-new, but instead of using the best previous values of already existing hyperparameters, the algorithm uses a Gaussian process fitted on the projected results of the previous HPO. This change had little effect on the performance of TGP.

Further, we performed the following minor revisions in response to the reviews:
* We have added missing details for our benchmark suite to Section 4 and Appendix A, and make it clearer that we use code and data from existing HPO benchmarks.
* We now list the adjustment type for each adjustment in our benchmarks in Table 1.
* We more clearly highlight why we include only-optimize-new and drop-unimportant, the two approaches inspired by actually-practiced manual strategies, in our evaluation.
* We now feature the combination of TGP (formerly T2PE) and best-first more prominently in the experiments section and show the per-benchmark performance in Appendix B.1.
* We improved the description for the comparison of GP (formerly TPE) across different seed ranges (Appendix E).
* We improved the description for the comparison of GP (formerly TPE) to random search and added a comparison of GP to TPE (Appendix F).
* We added an explanation for why we use the geometric mean to aggregate the speedups instead of the arithmetic mean.
* We added an explanation for why we use violin plots.

---

> ### Author Response · Authors · 2020-11-24
> **Paper Revisions After Replies**
>
> We performed the following major revisions in response to the replies to our initial response:
> * Based on a reviewer request, we added the TPE evaluation back in so we now look at two sets of baselines and transfer approaches.
> * We now provide a detailed discussion on the differences and contributions of HT-AA compared to existing work on hyperparameter transfer already in Section 2.
> * We improved the structure and explanations in Section 3 (Baseline Algorithms for HT-AA), especially for Transfer GP/TPE, and added Pseudocode for Transfer GP/TPE.
>
> We performed the following minor revisions in response to the replies to our initial response:
> * In some cases, we now show more than two approaches in a single plot.
> * We improved the exposition for the standardized improvement analysis (Appendix D).
> * We added per-benchmark plots for the standardized improvement analysis (Appendix D).

---

### Author Response · Authors · 2020-11-20
**Aspect of Introducing a New Problem**

We thank all reviewers for their helpful comments! We are glad that the new problem we proposed has been recognized as “valuable”, “interesting”, “clearly introduced”, and “quite relevant in practice” by the reviewers. We reply to the concerns of each reviewer separately in detail, and here, we comment on points that we see as relevant to all reviewers.

As reviewers are rarely in the situation where they review a paper with the main goal of drawing attention to a new problem setting, we would like to point out that the ICLR reviewing guidelines (https://iclr.cc/Conferences/2021/ReviewerGuide#step-by-step; 2.1) state that such papers, compared to papers that e.g., try to better address a known problem, require different considerations as to potential value and impact. We believe that a strong paper that introduces a problem (1) is centered around an interesting problem that is relevant in practice, (2) opens the gate for many new research opportunities, and (3) lays a solid foundation for further research on the problem. We believe that our paper fulfills all of these:

1. The reviewers clearly recognize hyperparameter transfer across adjustments (HT-AA) as an important problem that is relevant in practice.

2. In Section 6 (Related Work and Research Opportunities) we discuss some of the many research opportunities in extending the HT-AA framework, designing improved algorithms for HT-AA, and in applying the idea of automated knowledge transfers across developer adjustments to (meta) learning scenarios other than HPO. Additional ideas for future research directions are also given by the reviewers (e.g. HT-AA for ensemble learning). We believe the many opportunities for future research based on HT-AA clearly highlight the potential impact our paper can have.

3. To lay a solid foundation, the goal of our empirical study should not be to propose the best approach to this new algorithmic problem, or even to introduce exciting methodology, but (a) to provide code to researchers for cheap and hardware-independent benchmarks, and well-vetted baselines to base future work on, (b) to provide strong evidence for the advantage of automatically transferring knowledge about hyperparameters across adjustments, and (c) to evaluate how well actually-practiced manual strategies might work.
We deliver on all of these. In fact, the parts about baselines and evaluating actually-practiced strategies tie in with the question several reviewers asked of why we include the approaches only-optimize-new and drop-unimportant in our paper, even though they perform badly in the experiments. We do not propose only-optimize-new and drop-unimportant as new methods, but we rather include them as actually-practiced methods, alongside the straightforward best-first strategy and T2PE (now TGP). While best-first, T2PE/TGP, and the combination of these two work surprisingly well, only-optimize-new and drop-unimportant work horribly, even though dropping unimportant hyperparameters is a strategy practiced in a non-algorithmic fashion (e.g., in the seminal work on AlphaGO), and to only tune new hyperparameters is certainly widespread as well. We do not drop the evaluation of these two approaches as the negative results are evidence that actually-practiced manual approaches to the HT-AA problem perform worse than simply not doing any transfer at all. We changed the paper to more clearly highlight the points above.

---

### Decision · Program_Chairs · 2021-01-07
**Final Decision**

**Decision:**

Reject

**Comment:**

The paper has been actively discussed, both during and after the rebuttal phase. I enjoyed, and I am thankful for, the active communication that took place between the authors and the reviewers.

On the one hand, the reviewers agreed on several pros of the paper, e.g.,
* Clear, well presented manuscript
* The presentation of practically-relevant setting
* A work that fosters reproducible research (both BO data and algorithms are made available)
* Careful experiments

On the other hand, several important weaknesses were also outlined, e.g.,
* _Novelty_: While the authors claim they “introduce a practically relevant and fundamentally novel research problem”, existing commercial HPO solutions already mention, and propose solutions for, the very same problem, e.g., [AWS](https://docs.aws.amazon.com/sagemaker/latest/dg/automatic-model-tuning-warm-start.html) (section “Types of Warm Start Tuning Jobs”) and [Google cloud](https://cloud.google.com/blog/products/gcp/hyperparameter-tuning-on-google-cloud-platform-is-now-faster-and-smarter) (section “Learning from previous trials”). The reviewers all agreed on the fact that this down-weights the novelty aspect (claimed many times in the rebuttal and the manuscript): The paper formalizes an already existing framework rather than introducing it.
* In the light of the weakened "novelty" contribution (see above), the reviewers regretted the absence of a novel transfer method _tailored to HT-AA_, which would have certainly strengthened the submission.
* _“Dynamic range” of the benchmark_: It is difficult to evaluate the capacity of the benchmark to discriminate between different approaches (e.g., see new Fig. 3 showing the violin plot with all three methods for transfer, as suggested by Reviewer 1: the improvements over "best first" seem marginal at best). To better understand the benchmark, it would be nice to illustrate its “dynamic range” by exhibiting a more powerful method that would substantially improve over “best first”.

As illustrated by its scores, the paper is extremely borderline. Given the mixed perspectives of pros and cons, we decided with the reviewers to recommend the rejection of the paper.